# SelfBudgeter: Adaptive Token Allocation for Efficient LLM Reasoning

## Abstract

Recently, large reasoning models demonstrate exceptional performance on various tasks. However, reasoning models inefficiently over-process both trivial and complex queries, leading to resource waste and prolonged user latency. To address this challenge, we propose SelfBudgeter - a self-adaptive controllable reasoning strategy for efficient reasoning. Our approach adopts a dual-phase training paradigm: first, the model learns to pre-estimate the reasoning cost based on the difficulty of the query. Then, we introduce budget-guided GPRO for reinforcement learning, which effectively maintains accuracy while reducing output length. SelfBudgeter allows users to anticipate generation time and make informed decisions about continuing or interrupting the process. Furthermore, our method enables direct manipulation of reasoning length via pre-filling token budget. Experimental results demonstrate that SelfBudgeter can dynamically allocate budgets according to problem complexity, yielding an average response length compression of 61% for the 1.5B model on GSM8K, MATH500, and AIME2025, and 48% for the 7B model, while maintaining nearly undiminished accuracy.

## 1 Introduction

Recent large reasoning models, such as O1 (OpenAI, 2024), has shown remarkable performance in various complex reasoning tasks (DeepSeek-AI et al., 2025; Qwen, 2024). The primary success factor lies in the long chain of thought (CoT) process learned through reinforcement learning (RL), which allows the model to break down reasoning steps and scaling test-time compute (Snell et al., 2024; Luo et al., 2025b).

However, reasoning models tend to use overly long thought processes even for simple questions. This "overthinking" phenomenon leads to a waste of computational resources and excessive user waiting times (Chen et al., 2024; Sui et al., 2025). For example, when answering the simple questions such as "What is the answer of 2+3?", the QwQ-32B model provides 13 different solutions and generates 100 times more tokens than Qwen2.5-72B-Instruct model (Qwen et al., 2025).

Prior studies have explored various approaches to mitigate overthinking through response length control and computation routing. Existing methods mainly include: (1) Prompt-based approaches (Lee et al., 2025; Xu et al., 2025a) that implicitly guide length through instructions, (2) Integrated training strategies that teach models to adaptively determine reasoning steps via SFT (Munkhbat et al., 2025; Ma et al., 2025) or RL with length penalties (Aggarwal & Welleck, 2025; Arora & Zanette, 2025), and (3) Router-based (Aytes et al., 2025; Chuang et al., 2025) architectures employing classifiers to allocate computation paths. While achieving partial progress, these methods either lack precise length control, require additional computational overhead, or fail to explicitly output optimal reasoning lengths (Aggarwal & Welleck, 2025; Xu et al., 2025b).

We propose **SelfBudgeter** that enables reasoning models to (1) estimate the minimal token budget required for correct responses when users do not specify token constraints, and (2) generate responses of corresponding lengths while adhering to either self-estimated or user-defined token budgets. SelfBudgeter aims to mitigate the overthinking issue by predicting the minimal possible token budget, thereby significantly reducing user waiting time. As shown in Figure 1, SelfBudgeter can provide a relatively accurate token budget estimation before generating responses, users can precisely anticipate the waiting time and decide whether to wait for the full output or terminate early based on their needs.

Additionally, when specific requirements arise, users can pre-fill the token budget field to constrain the model's response within the given limit, thereby improving interaction efficiency.

Our training framework consists of two main stages. During the Cold-Start stage, we fine-tune the model to learn how to first output its estimated token budget within `<budget>` tags. Subsequently, in the RL training stage, we optimize SelfBudgeter using the GRPO algorithm. For this stage, we design a reward function that primarily focuses on three key aspects: (1) answer correctness, (2) minimal achievable token budget, and (3) consistency between response length and the allocated token budget.

We conduct full-parameter training of Deepseek-R1-Distill-Qwen-1.5B using SelfBudgeter and evaluate its performance on the GSM8K, MATH500 and AIME2025 datasets. Experimental results demonstrate that SelfBudgeter achieves an average response length compression of 61% with the 1.5B model, while maintaining nearly equivalent accuracy. Furthermore, on GSM8K and MATH500, SelfBudgeter simultaneously reduces response length while improving accuracy. SelfBudgeter also exhibits excellent capability in predicting output length and, when provided with pre-filled `<budget>` tags, consistently adheres to the specified token budget constraints. In addition, experiments on Deepseek-R1-Distill-Qwen-7B show an average compression of 48%, further validating the scalability of SelfBudgeter to larger model sizes.

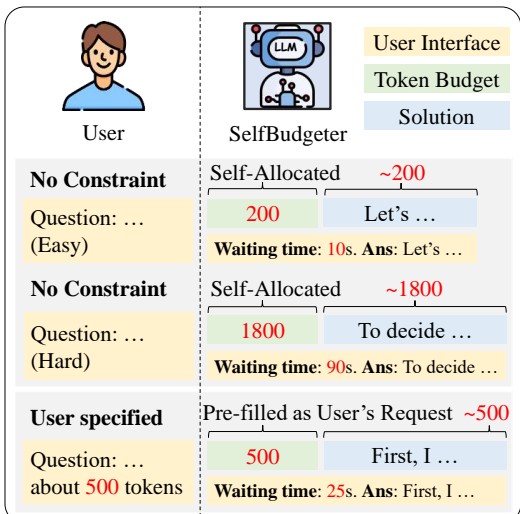

Figure 1: Overview of the SelfBudgeter. SelfBudgeter's responses comprise two sections: **Token Budget** and **Solution**. For unrestricted queries, it estimates tokens needed for the **Solution** based on problem complexity. When users specify requirements, it pre-fills the **Token Budget** accordingly. The **Solution** generation strictly adheres to **Token Budget** limits, whose numerical value indicates anticipated response duration.

## 2 RELATED WORK

**Overthinking in LLMs** The emergence of the reasoning models like O1, Deepseek-R1 and QwQ advanced complex problem-solving through RL-enhanced CoT (Wei et al., 2022; OpenAI, 2024; DeepSeek-AI et al., 2025; Qwen, 2024). However, researchers observed a tendency for reasoning models to overthink simple problems—expending unnecessary computational effort on trivial queries (Chen et al., 2024; Sui et al., 2025). Excessive long CoT may lead to a decrease in accuracy (Wu et al., 2025). Current solutions for overthinking mainly involve following three strategies. Prompt-based methods try to control response length by adding instructions in prompts, but cannot control the length accurately (Lee et al., 2025; Renze & Guven, 2024; Xu et al., 2025a; Nayab et al., 2024). Integated Training-based methods try to teach model decide the length by the difficulty of the problems. Supervised fine-tuning(SFT)-based methods collect the dataset with variable length (Munkhbat et al., 2025; Ma et al., 2025; Liu et al., 2024; Han et al., 2024; Kang et al., 2024; Xia et al., 2025; Yang et al., 2025b). RL-based methods incorporate length penalties into the reward function (Aggarwal & Welleck, 2025; Arora & Zanette, 2025; Luo et al., 2025a; Chen et al., 2025a; Chang et al., 2025; Xu et al., 2025b; Yang et al., 2025a). These methods fail to control the length as users' requirements. And Router-based methods train another model as a classifier (Aytes et al., 2025; Chuang et al., 2025; 2024; Ong et al., 2024; Pan et al., 2025). The classifier decide to route the query to fast models or reasoning models. However, an extra classifier means more computation resources are needed. Current methods either sacrifice precise control, require extra computation, or fail to bridge autonomous budget estimation with strict adherence.

**Token Budget** In addressing the issue of overthinking, a highly intuitive approach involves directly constraining the output length. CCoT (Nayab et al., 2024) attempt to achieve this by incorporating a word budget into the prompt, various approaches—including character, token, and step budgets (Lee

et al., 2025)—have been attempted by directly incorporating them into prompts, yet achieving precise control over the model's output behavior remains challenging. TALE (Han et al., 2024) introduce, for the first time, the concept of a token budget. TOPS (Yang et al., 2025b) attempt to enable the model to autonomously determine the required effort for solving a given task. However, both TALE and TOPS fail to explicitly guide the model to produce the optimal token budget. They also fail to effectively control the output length according to a given token budget. L1 (Aggarwal & Welleck, 2025) and Elastic Reasoning (Xu et al., 2025b) can more precisely control the output length under a given token budget, yet they fail to enable the model to autonomously estimate an appropriate response length. Our proposed method enables the model to autonomously estimate the optimal token budget and subsequently generate text in strict adherence to it.

## 3  METHOD

To minimize the overthinking problem in LLMs, we propose SelfBudgeter for efficient reasoning. Our method aims to enable the model to autonomously determine an appropriate token budget and generate responses of corresponding length while adhering to this budget. Although reasoning models may occasionally overthink simple problems, their response lengths generally increase with problem difficulty. This phenomenon demonstrates that the model possesses the capability to allocate token quantities reasonably based on problem complexity. Previous works such as L1 (Aggarwal & Welleck, 2025) and Elastic Reasoning (Xu et al., 2025b) have also demonstrated that models can generate responses of appropriate length according to a given token budget.

Therefore, we design SelfBudgeter, which employs a reward function to guide the model in: (1) learning an output format where it first predicts a token budget before generating the answer, (2) allocating appropriate token budgets based on its own capabilities and question difficulty, and (3) generating solutions with optimal length while ensuring answer accuracy.

### 3.1  SELFBUDGETER

SelfBudgeter is a concise and efficient method for automatic precise length controlled. We design the Precise Budget Control Reward (PreB Reward) to achieve precise control over length. The detailed introduction of PreB Reward can be found in Section 3.3. We employ GRPO algorithm to train the model in predicting appropriate token budgets based on problem difficulty and generating responses with lengths conforming to the specified budget.

Our reward function is formally defined as Formula 1:

$$
R(C, F, \ell, b, b_{\max}) = \begin{cases} r_f, & \text{if } F = 0, \\ P_B(b, b_{\max}) + \text{PreB}(s^W_{\min}, s^W_{\max}, \ell, b, \alpha, b^W_{\text{best}}), & \text{if } F = 1 \text{ and } C = 0, \\ P_B(b, b_{\max}) + \text{PreB}(s^C_{\min}, s^C_{\max}, \ell, b, \alpha, b^C_{\text{best}}), & \text{if } F = 1 \text{ and } C = 1. \end{cases} \quad (1)
$$

where

$$
b^C_{\text{best}} = (1 - \alpha) \cdot b, \quad b^W_{\text{best}} = (1 + \alpha) \cdot b \quad (2)
$$

The inputs and hyperparameters in the reward function are listed in Table 1. To ensure stable prediction of the token budget prior to response generation, any responses deviating from the prescribed format will be assigned the minimum reward score of $r_f$.

### 3.2  BUDGET PENALTY

To enable the model to learn token budget allocation, we introduce a budget penalty module defined by Formula 3. The model incurs a penalty $r_b$ when its estimated token budget exceeds the maximum acceptable budget $b_{\max}$. No penalty is applied when the estimated token budget remains within $b_{\max}$. A detailed introduction of $b_{\max}$ is presented in Section 4.2. Briefly stated, for a given question, $b_{\max}$ equals the response length if the base model can answer it correctly; otherwise, $b_{\max}$ is set to $\infty$.

$$
P_B(b, b_{\max}) = \begin{cases} 0, & \text{if } b \le b_{\max}, \\ r_b, & \text{else.} \end{cases} \quad (3)
$$

Table 1: Input and Hyperparameters (HPs) in the reward function

| Input | Description | HPs | Description |
|-------|-------------|-----|-------------|
| $C$ | Correctness for answer | $r_f$ | Penalty for format error |
| $F$ | Correctness for format | $s_{\min}^{W/C}$ | Minimum reward (wrong/correct) |
| $l$ | Response length | $s_{\max}^{W/C}$ | Maximum reward (wrong/correct) |
| $b$ | Model's budget | $\alpha$ | Tightness coefficient of budget |
| $b_{\max}$ | Maximum acceptable budget | $r_b$ | Penalty for excessive budget |

## 3.3 PRECISE BUDGET CONTROL REWARD

Inspired by the cosine reward (Chang et al., 2025), we propose the Precise Budget Control Reward (PreB Reward). While the cosine reward helps mitigate overthinking tendencies, it lacks precise control over output length, as it only constrains the upper bound of the response. To address this limitation, we introduce a tightness coefficient $\alpha$ to better align the response length with the specified token budget.

Given the inherent challenge for models to precisely comply with token budgets, we relax the length constraint to require only approximate adherence within $\alpha \cdot b$ around the target budget $b$. As shown in Formula 4, when the model's response length falls outside the specified range, the corresponding reward score plummets to its minimum value $s_{\min}$.

For incorrect responses, the function incentivizes longer reasoning chains (increasing length $\ell$) to encourage deeper analysis that might lead to correct conclusions. Conversely, for correct answers, the reward peaks at the minimally sufficient length $(1 - \alpha) \cdot b$ to prevent unnecessary computational overhead while maintaining accuracy. This explains why in Formula 2, the value of $b_{\text{best}}$ differs between correct and incorrect responses from the model. This dual mechanism promotes efficient reasoning by adaptively modulating response lengths based on answer correctness.

$$
\text{PreB}(s_{\min}, s_{\max}, \ell, b, \alpha, b_{\text{best}}) = \begin{cases} s_{\min}, & \text{if } \dfrac{|\ell - b|}{b} > \alpha, \\ s_{\min} + (s_{\max} - s_{\min}) \times \\ \quad \dfrac{1}{2}\left(1 + \cos\left(\pi \cdot \dfrac{|\ell - b_{\text{best}}|}{2\alpha b}\right)\right), & \text{else.} \end{cases} \tag{4}
$$

## 3.4 ACCURACY REWARD

To ensure the model's post-training accuracy does not degrade below its initial performance, we configure hyperparameters to guarantee that the minimum reward for correct responses always exceeds the maximum reward for incorrect responses. Specifically, our design ensures that: A correct response, which has a token budget exceeding $b_{\max}$ and receives the lowest budget following reward $s_{\min}^C$, will yield a higher total reward than an incorrect response that has a token budget within $b_{\max}$ and receives the highest budget following reward $s_{\max}^W$. This constraint is formally expressed as: $s_{\min}^C + r_b \geq s_{\max}^W$.

Overall, the core design of SelfBudgeter consists of three key modules: Budget Penalty, Preb Reward, and Accuracy Reward, which collectively balance length compression, correctness, and precise length control–ultimately delivering a better user experience.

## 4 EXPERIMENT

### 4.1 TRAINING TEMPLATE

The existing reasoning models utilize a pair of `<think></think>` tags to demarcate the thinking process from the final solution output. Building upon this format, we have further incorporated a token budget component.

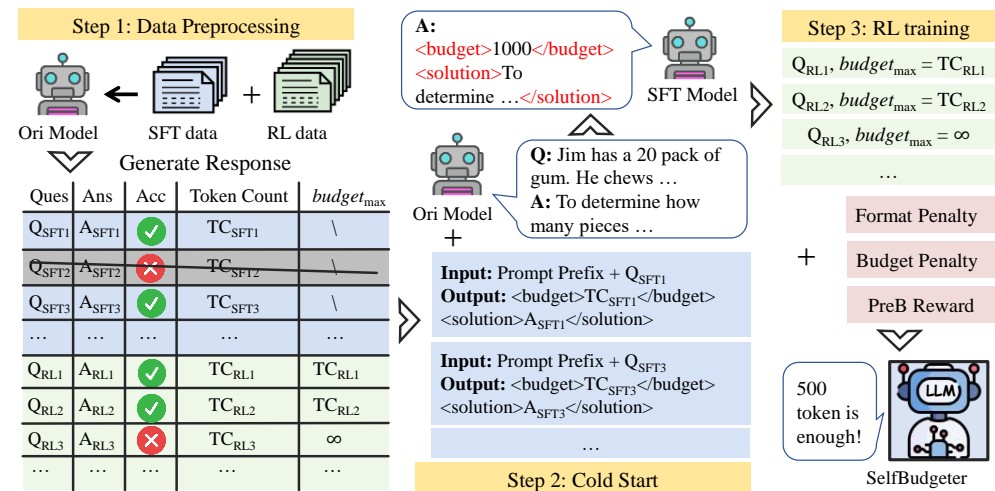

Figure 2: Overview of the SelfBudgeter training framework. The workflow consists of three key steps: (1) **Data preprocessing**: The initial model preprocesses the data to compute token budgets for subsequent training; (2) **Cold-start**: The model is fine-tuned to adopt the new output format; (3) **RL Training**: Through reward functions, the model learns to allocate token budgets and generate compliant outputs.

To enable the model to dynamically allocate token usage based on question difficulty, we design an output format as follows:

```
<budget>an integer</budget><solution>response</solution>
```

The format requires the model to first estimate the required token budget before providing the answer to the question. When no user constraint exists, the model autonomously predicts the token budget. When users specify a token limit, we pre-fill the <budget> field and let the model generate the <solution> within this constraint.

## 4.2 DATA PREPROCESSING

At this stage, we collect model's responses to the test questions used in both the cold-start and RL training phases, and then evaluate the correctness and length of the responses.

For the cold-start data, we retain the model's correct responses along with their lengths and discard incorrect answers to prevent reinforcing the model's memory of wrong responses.

For the RL training data, we calculate $budget_{\max}$ (for convenience, we will refer to it as $b_{\max}$ in the following sections) using Formula 5, representing the maximum acceptable token budget for a given question. When the model answers correctly, the correctness of the response indicates that the minimum token budget required for a correct answer does not exceed the current length. Therefore, we encourage the model to further compress the response length and set $b_{\max}$ to the current response length. When the model answers incorrectly, the relationship between the minimum token budget needed for correctness and the current length remains unclear, so any token budget is acceptable.

$$b_{\max} = \begin{cases} \text{response length,} & \text{if model answers correctly,} \\ \infty, & \text{else.} \end{cases} \tag{5}$$

## 4.3 COLD START

In our actual RL training process, we observe that requiring the model to simultaneously master multiple objectives - learning the new output format, providing appropriate token budgets, generating solutions of corresponding lengths according to the budget, while maintaining or improving accuracy

- proved excessively challenging. After extended training periods, the model often only succeeds in adopting the output format without achieving the other goals. Inspired by the Deepseek-R1 training methodology, we introduce a cold-start phase to accelerate training and enable the model to first learn the new output format before proceeding to more complex tasks. The overall training framework is illustrated in Figure 2.

To prevent the model from losing its original reasoning capability during the cold-start phase, fine-tuning must be performed using either the model's own generated responses or datasets containing long CoT responses. In our approach, we pre-populate the `<budget>` section with token counts obtained during the preprocessing stage. The `<solution>` section is filled with the model's generated responses. And the instruction prefix we prepend to each question can be found in Appendix B .

### 4.4    EXPERIMENT SETTINGS

We conduct experiments on the DeepSeek-R1-Distill-Qwen-1.5B (R1-1.5B) model. We reproduce L1-Max using R1-1.5B, and select R1-1.5B and L1-Max as baseline methods for comparative evaluation against SelfBudgeter. In addition, we extend our experiments to the larger DeepSeek-R1-Distill-Qwen-7B (R1-7B) model. For more comprehensive comparison, we also include E1-Math-1.5B, R1-7B, Eurus-2-7B-PRIME (Cui et al., 2025), and Qwen-2.5-7B-Simple-RL (Shao et al., 2024) as additional baselines.

During the cold-start phase, we employ three datasets of varying difficulty—GSM8K (Cobbe et al., 2021), MATH (Hendrycks et al., 2021), and s1k-1.1 (Muennighoff et al., 2025)—to help the model learn the new output format while producing token budgets with diverse distributions. The s1k-1.1 dataset contains 1,000 challenging mathematical problems with long reasoning chains generated by DeepSeek-R1, which support both reasoning ability and format adaptation. For GSM8K and MATH, we select 1,500 training samples each that the model can answer correctly. For s1k-1.1, we directly use the native responses and compute the corresponding token counts with the model's tokenizer to populate our designed template; in total, we retain 630 problems that DeepSeek-R1 answered correctly. This yields a training set of 3,630 samples. Following the preprocessing protocol in Sections 4.2 and 4.3, we fine-tune the model for one epoch. Throughout data collection and training, the model's temperature is consistently set to 0.6.

During the reinforcement learning phase, we use STILL-3-Preview-RL-Data (Chen et al., 2025b) dataset. It also serves as the training dataset for reproducing L1-max. This dataset collects 30K high-quality samples based on the MATH (Hendrycks et al., 2021), NuminaMathCoT (LI et al., 2024), and AIME 1983-2023 (Veeraboina, 2023) datasets. It includes problems of varying difficulty levels, which also helps the model learn to allocate token counts adaptively based on difficulty. As described in Section 4.2, we compute the maximum acceptable budget ($b_{\max}$) based on the model's responses, then train the model for 3 epochs on this dataset. More detailed information can be found in Appendix A.

### 4.5    MAIN RESULTS

Table 2 presents a comprehensive comparison of model performance on the GSM8K, MATH500, and AIME2025 test sets, evaluated in terms of accuracy (Acc) and average response length (Len). The table contrasts baseline models with different variants of the SelfBudgeter framework across varying model scales. For clarity, the best performance is highlighted in bold, while the second-best performance is indicated with underline. It is worth noting that token limits for L1 are explicitly specified through prompt templates, whereas those for E1 are enforced via hard truncation. In contrast, SelfBudgeter autonomously estimates its token constraints during inference. All reported results are averaged over three runs with different random seeds.

**Baseline Comparison**    Although the Deepseek-R1-Distill-Qwen-1.5B baseline demonstrates strong accuracy, it requires substantially longer responses. On GSM8K, our method improves accuracy by 11.01 percentage points while compressing response length to 43% of the original. On MATH500, it achieves a 3.54-point accuracy gain with response length reduced to 44%. On AIME2025, our approach compresses response length to 30% of the original while maintaining comparable accuracy. In contrast, although L1 and E1 attain stronger compression on certain datasets, they incur larger

Table 2: Performance comparison on GSM8K, MATH500, and AIME2025. Accuracy (Acc) is reported in percentage, and length (Len) in tokens.

| Models | GSM8K | | MATH500 | | AIME2025 | |
|---|---|---|---|---|---|---|
| | Acc | Len | Acc | Len | Acc | Len |
| DeepSeek-R1-Distill-Qwen-1.5B | 73.09 | 2865.08 | 74.93 | 5327.12 | **22.22** | 14444.03 |
| E1-Math-1.5B(0.5K,1K) | 60.20 | 1205.21 | 35.53 | 1499.54 | 4.44 | 3008.44 |
| E1-Math-1.5B(4K,1K) | 72.10 | 1299.62 | 72.47 | 2088.44 | 21.11 | 5578.13 |
| L1-Max(3600) | 79.56 | 571.72 | 76.73 | 1753.42 | 17.88 | 5213.89 |
| SelfBudgeter-1.5B | **84.10** | 1231.79 | **78.47** | 2326.85 | 21.11 | 4288.10 |
| DeepSeek-R1-Distill-Qwen-7B | 87.09 | 1918.21 | 86.73 | 5387.19 | 28.89 | 22158.79 |
| Eurus-2-7B-PRIME | **90.98** | 302.72 | 79.73 | 582.58 | 15.56 | 1254.52 |
| Qwen-2.5-7B-Simple-RL | 75.94 | 519.07 | 61.13 | 823.89 | 6.67 | 1429.94 |
| SelfBudgeter-7B | 90.30 | 991.13 | **86.87** | 2666.58 | **30.00** | 12241.84 |

accuracy losses—L1 performs poorly on the challenging AIME2025 benchmark, while E1 suffers more pronounced accuracy degradation on the simpler GSM8K and MATH500 datasets.

In addition, Table 2 highlights that SelfBudgeter consistently strikes a better balance between accuracy and response length than existing baselines. Unlike L1, which enforces explicit length limits but collapses on AIME2025, or E1, which relies on hard truncation and severely harms accuracy, SelfBudgeter autonomously learns effective token budgeting. As a result, it achieves the best or second-best accuracy across all datasets while simultaneously reducing response length substantially.

Beyond its effectiveness at the 1.5B scale, our method also delivers efficient reasoning with larger models. SelfBudgeter-7B achieves the highest accuracy on MATH500 and AIME2025, and the second-best accuracy on GSM8K—only 0.68 points lower than the best-performing model. Meanwhile, SelfBudgeter-7B attains an average compression ratio of 48%, further demonstrating the generality of our approach and its effectiveness at larger model scales. Compared with Eurus-2-7B-PRIME, which excels only on GSM8K but falls behind on harder reasoning tasks, and Qwen-2.5-7B-Simple-RL, which underperforms across all benchmarks, SelfBudgeter exhibits robust gains across datasets of varying difficulty.

## 4.6 DYNAMIC ALPHA SCHEDULE

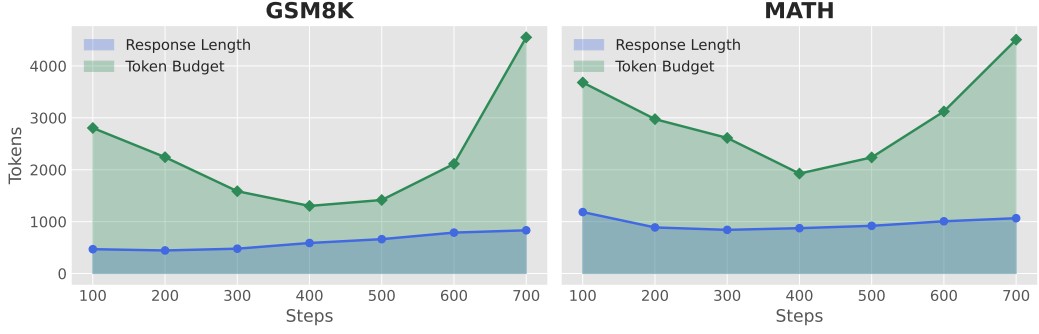

Figure 3: Response length and token budget on GSM8K and MATH benchmarks across training steps with $\alpha = 0.5$. The curves show how the average response length (solid circles) and allocated token budget (solid diamonds) evolve during training.

In SelfBudgeter, $\alpha$ serves as a critical hyperparameter. As shown in Figure 3, we observe that using a fixed and relatively loose $\alpha$ can lead to *reward hacking*: once the model learns to align the budget with the actual response length, it tends to inflate the predicted budget during later training stages, pushing the output length toward the lower bound of the acceptable range to obtain higher PreB

scores. Conversely, when $\alpha$ is fixed but relatively tight, the token budget quickly collapses to the response length, which hinders the model from learning an optimal budgeting strategy. To address these issues, we introduce a *dynamic alpha schedule*, where $\alpha$ is linearly decreased over training steps. This gradually tightens the tolerance range for acceptable response lengths and encourages closer convergence between the predicted budget and the actual output length. Consequently, the optimal $\alpha$ is not static but evolves throughout the training process.

Formally, the dynamic $\alpha$ is defined by a linear schedule:

$$\alpha_{\text{now}} = \alpha_{\text{start}} - (\alpha_{\text{start}} - \alpha_{\text{end}}) \cdot \frac{\text{step}_{\text{now}}}{\text{Total steps}}. \tag{6}$$

This schedule only requires specifying the starting and ending values of $\alpha$ (i.e., $\alpha_{\text{start}}$ and $\alpha_{\text{end}}$), which are set to $6.0$ and $0.1$, respectively.

## 5 DISCUSSION

In the Analysis section, we systematically examine SelfBudgeter's adaptive computation allocation mechanism through two pivotal aspects: its ability to dynamically adjust budgets according to problem complexity, and compliance with token constraints while preserving response quality. This holistic evaluation reveals fundamental insights into how adaptive language models negotiate computational efficiency with task requirements, informing both theoretical understanding and practical deployment considerations.

### 5.1 ADAPTIVE BUDGET ALLOCATION

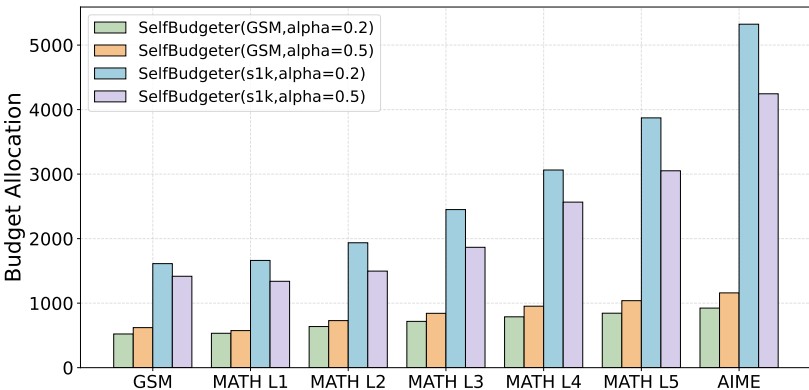

Figure 4: Token budget allocation patterns across problem difficulty levels for four SelfBudgeter-1.5B configurations (initialized on GSM8K/s1k with $\alpha$=0.2/0.5). All variants exhibit monotonic budget escalation with increasing task complexity (GSM8K, MATH Level 1-5, AIME2024), confirming robust cross-configuration alignment between computational investment and intrinsic problem difficulty.

To investigate SelfBudgeter's capacity for difficulty-aware budget allocation, we conduct empirical evaluations across three mathematical reasoning benchmarks with inherent complexity gradients: GSM8K, MATH, and AIME 2024. Our experimental framework systematically evaluates four architectural variants combining cold-start initialization strategies (GSM8K vs. s1k) with $\alpha$ hyperparameter values (0.2 vs. 0.5).

Figure 4 shows a consistent positive correlation between problem complexity and allocated token budgets across all model variants, demonstrating SelfBudgeter's ability to scale computation with task difficulty. The near-linear allocation across difficulty tiers highlights its emergent capacity for intrinsic difficulty estimation, while the minimal variance across configurations indicates robust and generalized learning of task-complexity metrics rather than configuration-specific artifacts.

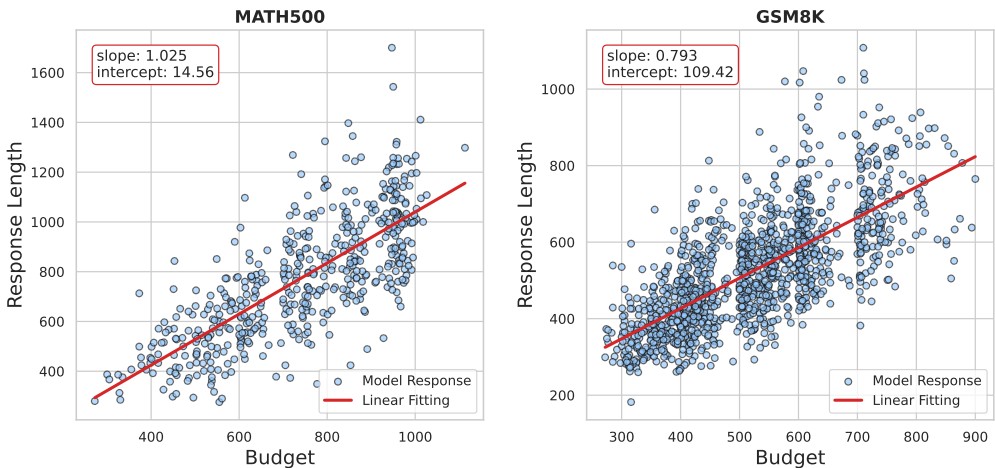

Figure 5: Linear regression analysis of token budget prediction versus actual response length for SelfBudgeter on the MATH500 dataset(left) and GSM8K test set(right). The figure demonstrates SelfBudgeter with GSM initialization and hyperparameter $\alpha = 0.2$.

## 5.2 BUDGET-CONSTRAINED GENERATION

To systematically evaluate the generation capability of SelfBudgeter under budget constraints, this study employs linear regression modeling to quantitatively analyze the mapping relationship between predicted token budgets and actual response lengths. We conduct a quantitative analysis on the MATH500 dataset and GSM8K test set using linear regression to investigate the mapping between predicted budgets and actual response lengths (as shown in the Figure 5). On MATH500 dataset, the least squares fitting yields a slope of 1.025 (95% CI [0.9466, 1.1042]). And on GSM8K test set, the least squares fitting yields a slope of 0.793 (95% CI [0.7512, 0.8354]). The slope coefficient approaching unity validates the efficacy of the budget control mechanism, indicating that each 1-token increase in the predicted budget corresponds to an average increase of about 1-token in output.

Quantitative results demonstrate that 96% of generated responses exhibit relative deviations $\leq 50\%$ from the target token budget, with 65.40% achieving tighter deviations $\leq 20\%$. Extended experiments on full benchmark datasets reveal that 97.65% (GSM8K) and 95.82% (MATH) of samples satisfy the $\leq 50\%$ relative deviation constraint. Notably, the model's budget adherence is influenced by the cold-start dataset and hyperparameter $\alpha$. The optimized SelfBudgeter configuration (initialized with GSM8K and $\alpha = 0.2$), which balances generation quality and budget compliance, is reported here as the best-performing variant.

We further validate SelfBudgeter's adherence to *user-defined* token budgets through controlled experiments. The results indicate that the actual generated length follows a linear functional relationship with user-defined budgets, demonstrating robust alignment even under explicit external constraints. Details are provided in Appendix C.

## 6 CONCLUSION

We propose the SelfBudgeter framework, which autonomously predicts required token budgets for reasoning while effectively adhering to self-imposed constraints, successfully optimizing the accuracy-response length trade-off. By leveraging SelfBudgeter's token budget predictions, users can anticipate total inference duration in advance, significantly enhancing user experience. In resource-efficient reasoning, SelfBudgeter demonstrates performance comparable to several existing methods, highlighting its potential for deployment in resource-constrained environments. Additionally, output length can be dynamically regulated through transformation functions when required. SelfBudgeter paves a promising pathway toward more efficient, controllable, and user-friendly reasoning models.

ETHICS STATEMENT

This work uses only publicly available datasets under their original licenses, and does not involve human subjects, private data, or personally identifiable information. Our contributions are methodological, focusing on improving reasoning efficiency, and do not amplify risks of harmful or biased content. We declare no conflicts of interest or ethical concerns, and we have complied with the ICLR Code of Ethics throughout the research and submission process. Additional details regarding the use of large language models (LLMs) are provided in Appendix F.

REPRODUCIBILITY STATEMENT

We have made extensive efforts to ensure the reproducibility of our work. The datasets used in our experiments are publicly available. Detailed descriptions of data preprocessing, training settings, and evaluation protocols are provided in Section 4, with additional implementation details and hyperparameters included in the appendix. We will release anonymous source code and scripts for training and evaluation as supplementary material.

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

# A    TRAINING DETAILS

## A.1    EXPERIMENTAL ENVIRONMENTS

Our server is equipped with two 80GB A100 GPUs and two 45GB A40 GPUs. We conducted fine-tuning experiments and inference tests on the two A40 GPUs, while the GRPO training was performed on the two A100 GPUs.

## A.2    PARAMETER SETTINGS

In the fine-tuning training during the cold-start phase, our parameter settings are configured as follows. The sequence length is capped at 16,384, with a per-device training and evaluation batch size of 1, while gradient accumulation (2 steps) is employed to alleviate GPU memory constraints. A cosine learning rate scheduler is adopted with a 10% warm-up ratio and a base learning rate of 5e-5. The model is trained for 1 epoch, with 10% of the training set allocated for validation. The model checkpoints are saved and evaluated every 500 steps, and the best-performing checkpoint is retained.

In the GRPO (Global Reward Policy Optimization) training, our parameter configuration is set as follows. The training and validation batch sizes are set to 128 and 1,250, respectively, with maximum prompt and response lengths of 1,024 and 32,000 tokens. The Actor model employs a learning rate of 1e-6, dynamic batching (up to 24K tokens per GPU), and a KL divergence loss (coefficient 0.001), with gradient checkpointing and FSDP (Fully Sharded Data Parallel) distributed training enabled (parameter offloading disabled). During the Rollout phase, the vLLM inference engine is utilized with tensor parallelism (TP=2) and 80% GPU memory utilization, generating 5 responses per round. Global settings include 3 training epochs, a checkpoint-saving interval of 50 steps, and a KL control coefficient of 0.001, executed on a single node with dual GPUs. And key hyperparameters involved in the reward function are specified in Table 3.

Table 3: Hyperparameters Settings

| Parameters | $C = 0$ | $C = 1$ | Parameters | Value |
|:---:|:---:|:---:|:---:|:---:|
| $s_{\min}$ | -0.5 | 0.5 | $r_f$ | -1 |
| $s_{\max}$ | 0 | 1 | $r_b$ | -0.4 |

For the GSM-initialized SelfBudgeter, we select the checkpoint after 699 training steps when alpha was set to 0.2, and the checkpoint after 575 steps when alpha was 0.5. For the s1k-initialized SelfBudgeter, we choose the checkpoint after 475 training steps with alpha=0.2, and the checkpoint after 500 steps with alpha=0.5. For L1-Max, we choose the checkpoint after 280 training steps.

# B    COLD-START DATA SELECTION

> **Prompt Template**
>
> Answer the given question. You should first estimate the total number of tokens you will need to answer this question based on its difficulty. Then you think about the reasoning process in the mind and provide the user with the answer. The token budget and whole solution are enclosed within <budget></budget> and <solution> </solution> tags, respectively, i.e., <budget> token budget here, just an integer </budget><solution> solution here, please output the final answer within \boxed{} </solution>.
> Question:

Figure 6: The prompt template used in the cold-start stage.

The choice of initialization data substantially impacts model performance. SelfBudgeters initialized with the **s1k** dataset outperform their GSM-initialized SelfBudgeters by 8.82–10.72 percentage points

Table 4: Model performance comparison on GSM8K and MATH test sets, showing accuracy (Acc/%), average response length (Len/tokens) and matching rate between token limits and response length (Mat/%). The SelfBudgeter variants with different cold-start data and $\alpha$ parameters are contrasted with baseline models.

| Model | GSM8K | | | MATH | | |
|---|---|---|---|---|---|---|
| | Acc↑ | Len↓ | Mat↑ | Acc↑ | Len↓ | Mat↑ |
| Cold Start (GSM) | 71.95 | 1003.79 | 85.82 | 64.74 | 3043.29 | 41.16 |
| SelfBudgeter (GSM, $\alpha = 0.2$) | 76.27 | 523.77 | **97.65** | 63.46 | 779.54 | 95.82 |
| SelfBudgeter (GSM, $\alpha = 0.5$) | 74.68 | 520.82 | 96.97 | 63.78 | 777.80 | **96.66** |
| Cold Start (s1k) | 82.49 | 1983.29 | 21.76 | 76.64 | 4001.29 | 23.28 |
| SelfBudgeter (s1k, $\alpha = 0.2$) | **81.50** | 662.08 | 70.74 | **74.18** | 919.27 | 78.36 |
| SelfBudgeter (s1k, $\alpha = 0.5$) | 80.44 | 719.36 | 71.19 | 72.60 | 1022.99 | 79.76 |

on MATH (74.18% vs. 63.46% for $\alpha = 0.2$) and 5.23–5.76 percentage points on GSM8K (80.44% vs. 74.68% for $\alpha = 0.5$). While SelfBudgeters with GSM-initialized exhibit lower accuracy, they generate significantly more concise responses compared to s1k-initialized SelfBudgeters. Specifically, GSM-initialized SelfBudgeters reduces response length by approximately 15–24% on MATH and achieves 21–28% length reduction on GSM8K. This performance gap highlights the importance of high-quality initialization for the budgeting mechanism.

As shown in Table 4, significant performance variations exist between models fine-tuned with different cold-start datasets. The s1k-fine-tuned model demonstrates superior accuracy over the GSM-fine-tuned counterpart, achieving 10.54% and 11.90% higher accuracy on GSM8K and MATH respectively. However, this comes at the cost of substantially longer responses, with the s1k model generating 97.58% and 31.48% lengthier outputs on GSM8K and MATH. This discrepancy stems from the s1k dataset's responses being generated by Deepseek-R1, which produces higher-quality outputs than those self-generated by Deepseek-R1-Distill-Qwen-1.5B. Additionally, the s1k dataset's average length of 7,677.43 tokens (we only retained correct responses under 16,000 tokens) vastly exceeds GSM8K's 837.14 tokens, explaining the dramatic difference in response lengths after fine-tuning. These factors substantially influence SelfBudgeter's final performance, as evidenced by: (1) SelfBudgeter's accuracy closely mirroring that of its fine-tuned base model, and (2) the response length relationships and matching rate relationships between different SelfBudgeter variants remaining consistent with their respective cold-start models.

## C PREFILLED TOKEN BUDGET FOLLOWING

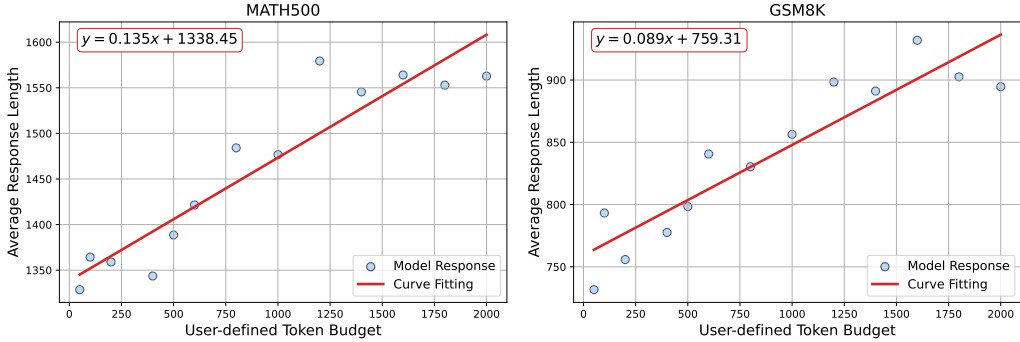

Figure 7: Relationship between user-defined token budgets and SelfBudgeter average response lengths with curve fitting analysis.

To systematically evaluate model performance under user-defined token budget constraints, we conduct quantitative analysis using SelfBudgeter with GSM initialization and hyperparameter $\alpha = 0.2$

on both MATH500 dataset and GSM8K test set. In the experimental design, fixed token budgets were pre-filled in the <budget> field of training templates, with empirical results obtained by measuring average generated response lengths. We evaluated SelfBudgeter's performance with user-defined token budgets ranging from 50 to 2000 (specifically: 50, 100, 200, 400, 500, 600, 800, 1000, 1200, 1400, 1600, 1800, and 2000), as shown in the Figure 7.

Regression intercepts effectively reflect problem complexity, where GSM8K's simpler questions yield significantly smaller intercepts. Despite a moderate slope, SelfBudgeter demonstrates robust budget adaptability, maintaining a stable positive correlation between user-defined budgets and output lengths. This linear relationship enables deterministic length control through derived transformation functions.

## D  CASE STUDY

To examine how SelfBudgeter's length compression affects reasoning quality, we compare the GSM8K outputs of R1-1.5B and SelfBudgeter-1.5B, focusing on cases where the compressed output is shorter than 70% of the original. Among the 381 samples where both models answer correctly, 96.06% of the reductions arise from eliminating conversational filler (e.g., "Alright, so I've got this problem here," "Hmm, okay," "Let me check"), while only 3.94% reflect genuinely more efficient problem-solving strategies. This suggests that SelfBudgeter primarily shortens CoTs by removing non-essential verbalization without altering the underlying logic.

We further analyze instances where length is reduced to under 70% and the two models diverge in correctness. In 13.12% of these cases (173 examples), the shorter reasoning chain produces a correct answer where the baseline fails, indicating that compression can improve reasoning by avoiding distractions or redundant steps. However, in 1.52% of cases (20 examples), the tightened budget omits crucial intermediate steps, causing the model to switch from a correct to an incorrect answer. Together, these results show that compression is generally benign or even beneficial, though excessively restrictive budgets can harm solution completeness. We also provide representative examples for each category: baseline wrong → compressed correct, baseline correct → compressed wrong, and both correct.

---

**Case 1: Wrong to Correct**

**Question**

Andrew travels from New Jersey to Rochester by bus, which takes 6 days. Traveling by car takes half as many days. If he goes by bus and returns by car, how many days will he travel?

**R1-1.5B's Answer**

**Answer**

The number of days Andrew will travel is $\boxed{12}$.

**Step-by-step explanation**

First, determine the number of days to travel by bus from New Jersey to Rochester. Since it takes 6 days by bus, the return trip by car will take half as many days, which is 3 days.

Next, add the days for the trip to and return from New Jersey to Rochester: 6 days for the bus to New York City plus 3 days for the car back to New Jersey.

Finally, sum up these two durations to find the total travel time.

To determine the total number of days Andrew will travel, we need to consider both the trip to New York City by bus and the return trip by car.

**Step 1: Calculate the travel time by bus**

It takes **6 days** to travel from New Jersey to New York City by bus.

**Step 2: Calculate the return travel time by car**

The return trip by car takes **half as many days** as the bus trip to New York City. Therefore, the travel time by car is:

$$\text{Car travel time} = \frac{6}{2} = 3 \text{ days}$$

**Step 3: Calculate the total travel time**

Add the travel time by bus and the return travel time by car:

---

$$\text{Total travel time} = 6 \text{ days} + 3 \text{ days} = 9 \text{ days}$$

However, there's a correction needed. The problem states that Andrew plans to go to New York City by bus and return by car. This implies that the return journey is by car, which we have already calculated as 3 days.
Therefore, the total travel time is:

$$\text{Total travel time} = 6 \text{ days} + 3 \text{ days} = 9 \text{ days}$$

However, upon re-evaluation, the initial calculation might have been misinterpreted. Let's correct it:

**Correct Step 2: Calculate the return travel time by car**
The return trip by car takes:

$$\text{Car travel time} = \frac{6}{2} = 3 \text{ days}$$

**Step 3: Calculate the total travel time**

$$\text{Total travel time} = 6 + 3 = 9 \text{ days}$$

The correct total travel time is **9 days**.
However, considering the initial answer provided was $\boxed{12}$, the correct interpretation might be:
Let $x$ be the number of days to travel by bus to New York City.
Return trip by car takes $x/2$.

$$x + \frac{x}{2} = 6 + 3 = 9 \text{ days}$$

But to match the initial answer, we assume:

$$\boxed{12}$$

**SelfBudgeter-1.5B's Answer**

Okay, let's see. Andrew is planning a road trip from New Jersey to Rochester. The problem says it takes 6 days to travel by bus. Then it mentions that it takes half as many days to travel by car from New Jersey to Rochester. Hmm, so first, I need to figure out how many days it takes to get from New Jersey to Rochester by car. The problem says half as many days as by bus. Since the bus trip takes 6 days, the car trip should take half of that, right? Let me calculate that.
Half of 6 days is 3 days. So, the car trip from New Jersey to Rochester takes 3 days. Now, the question is asking how many days Andrew will travel if he plans to go to Rochester by bus and return by car. So, he's going there by bus and coming back by car. That means he'll be traveling from New Jersey to Rochester by bus, which takes 6 days, and then returning from Rochester to New Jersey by car, which takes 3 days. To find the total travel days, I need to add the days for the trip to Rochester and the days for the return trip.
So, 6 days (bus) plus 3 days (car) equals 9 days total. Let me just double-check to make sure I didn't miss anything. The problem says "how many days will he travel if he plans to go to Rochester by bus and return by car." So, yes, that's the total time for both trips. The key steps were figuring out the car trip time by taking half of the bus trip time and then adding both trips together. I think that's correct.
Wait, let me make sure about the wording. It says "it takes half as many days to travel by car from New Jersey to Rochester." So, if the bus takes 6 days, the car takes half of that, which is 3 days. Then, going to Rochester by bus is 6 days, and returning by car is 3 days. So, $6 + 3 = 9$ days total. Yeah, that seems right. I don't think I made any mistakes there.
Just to be thorough, let's consider if there's any possibility of overlap or different travel times, but the problem doesn't mention anything about stops, rest, or different routes, so I think it's a straightforward addition of the two travel times. Therefore, the total travel days are 9 days.
**Final Answer**
The total number of days Andrew will travel is $\boxed{9}$.

## Case 2: Correct to Wrong

**Question**

There are 4 snails in one aquarium and 32 snails in another aquarium. The difference between the number of snails in the two aquariums is twice the amount of fish in both aquariums. If both aquariums have the same number of fish in them, how many fish are there in each aquarium?

**R1-1.5B's Answer**

**Note: The problem might be tricky and require careful setup of equations.**

Okay, so I have this problem here about snails in two aquariums. Let me try to figure it out step by step. Sometimes these word problems can be a bit confusing at first, but if I break them down, they should make sense.

Alright, the problem says: There are 4 snails in one aquarium and 32 snails in another aquarium. The difference between the number of snails in the two aquariums is twice the amount of fish in both aquariums. If both aquariums have the same number of fish in them, how many fish are there in each aquarium?

Hmm, so let me parse this. First, there are two aquariums. One has 4 snails, the other has 32 snails. So, the number of snails in the first aquarium is 4, and in the second one is 32.

Now, the next part says: "The difference between the number of snails in the two aquariums is twice the amount of fish in both aquariums."

Wait, that's a bit of a mouthful. Let me try to translate that into math. The difference between the snails in the two aquariums is $32 - 4$, which is 28 snails. So, the difference is 28.

Then, it says that this difference (28) is twice the amount of fish in both aquariums. Hmm, so twice the amount of fish in both aquariums. So, first, I need to find the total number of fish in both aquariums. Let me denote the number of fish in the first aquarium as $F_1$ and the second as $F_2$.

So, the difference in snails is 28, and that's equal to twice the total number of fish in both aquariums. So, equation-wise, that would be:

$$28 = 2(F_1 + F_2)$$

So, that's equation one.

The problem also says, "If both aquariums have the same number of fish in them." So, that means $F_1 = F_2$.

Oh, that's a key point. So, if both aquariums have the same number of fish, then $F_1 = F_2$. Let's denote this common number as $F$. So, $F_1 = F_2 = F$.

So, substituting that into our first equation, we have:

$$28 = 2(F + F)$$

Which simplifies to:

$$28 = 2(2F)$$

Which is:

$$28 = 4F$$

So, solving for $F$:

$$F = \frac{28}{4}$$

$$F = 7$$

Wait, so each aquarium has 7 fish? Let me check.

So, if each aquarium has 7 fish, then the total number of fish is $7 + 7 = 14$.

The difference in the number of snails is $32 - 4 = 28$. And twice the total number of fish is $2 \times 14 = 28$. So, 28 equals 28. That checks out.

But wait a second, hold on. The problem says, "the difference between the number of snails in the two aquariums is twice the amount of fish in both aquariums." So, does this mean that

the difference is equal to twice the total number of fish? Or is it twice each of the number of fish in the aquariums? Hmm.

Let me re-read the problem to make sure.

"The difference between the number of snails in the two aquariums is twice the amount of fish in both aquariums."

So, it's the difference in snails is equal to twice the amount of fish in both aquariums. So, maybe it's $2(F_1 + F_2)$, which is what I did earlier.

But if it's twice the amount of fish in both aquariums, maybe it's $2F_1 + 2F_2$? Hmm, no, that doesn't make sense because that would be $2(F_1 + F_2)$. Wait, $2(F_1 + F_2)$ is 28, which is the difference in the number of snails.

Wait, now I'm getting confused.

Let me go back to the problem statement again.

"There are 4 snails in one aquarium and 32 snails in another aquarium. The difference between the number of snails in the two aquariums is twice the amount of fish in both aquariums."

So, the difference in snails is 32 − 4 = 28.

This difference is equal to twice the amount of fish in both aquariums.

So, the difference in snails (28) is equal to 2 × (amount of fish in both aquariums). So, 28 = $2(F_1 + F_2)$. That seems to be the correct interpretation.

But in that case, I think my original solution is correct.

So, total fish in both aquariums is 14, so each aquarium has 7 fish.

But let me make sure I'm not misinterpreting the problem. If it were $2F_1 + 2F_2$, that would be $2(F_1 + F_2)$, which is the same as before.

So: 28 = 2(F1 + F2) => F1 + F2 = 14 => each aquarium has 7 fish.

But let me think again, is this the correct interpretation?

Alternatively, maybe the problem is saying the difference is twice the amount of fish in each aquarium? So, $2F_1$ and $2F_2$, but that wouldn't make much sense, because the difference is a single value.

Thus:

$$28 = 2(F_1 + F_2)$$

Still seems right.

Hence, each aquarium has 7 fish.

Let me check consistency: Total fish = 14 Twice total fish = 28 Difference in snails = 28 → matches.

Alternatively, if each aquarium had 14 fish:

Total fish = 28 Twice total fish = 56 But the snail difference is only 28 → mismatch.

So that interpretation fails.

Thus:

$$28 = 2(F_1 + F_2) \Rightarrow F_1 + F_2 = 14$$

Given equal fish:

$$F_1 = F_2 = F \Rightarrow 2F = 14 \Rightarrow F = 7.$$

Therefore, each aquarium has 7 fish.

**Final Answer:** The number of fish in each aquarium is $\boxed{7}$.

Given that there are 4 snails in one aquarium and 32 snails in another, the difference in the number of snails between the two aquariums is 32 − 4 = 28. This difference is twice the amount of fish in both aquariums. Let $F_1$ and $F_2$ be the number of fish in the first and second aquariums, respectively. Since both aquariums have the same number of fish, we denote this common number as $F$. Therefore, $F_1 = F_2 = F$.

The equation given by the problem is:

$$28 = 2(F_1 + F_2)$$

Substituting:

$$28 = 2(F + F) = 4F$$

Thus:

$$F = 7$$

Thus, the number of fish in each aquarium is $\boxed{7}$.

### SelfBudgeter-1.5B's Answer

Alright, so I need to figure out how many fish are in each aquarium. Let me start by understanding the problem again.

There are two aquariums: one with 4 snails and another with 32 snails. The difference between the number of snails in the two aquariums is twice the amount of fish in both aquariums. Also, both aquariums have the same number of fish. I need to find out how many fish are in each aquarium.

First, let's break down the information given:

1. Number of snails in the first aquarium: 4  2. Number of snails in the second aquarium: 32 3. The difference between the number of snails in the two aquariums is twice the amount of fish in both aquariums. 4. Both aquariums have the same number of fish.

Wait, the problem says "the difference between the number of snails in the two aquariums is twice the amount of fish in both aquariums." Hmm, that's a bit confusing. Let me parse that again.

The difference between the number of snails in the two aquariums is twice the amount of fish in both aquariums. So, if I let $F$ be the number of fish in each aquarium, then the total number of fish in both aquariums is $2F$. The difference between the number of snails in the two aquariums is $32 - 4 = 28$. According to the problem, this difference (28) is twice the amount of fish in both aquariums ($2F$). So:

$$28 = 2F$$

Solving for $F$, we get:

$$F = 14.$$

So, each aquarium has 14 fish. Let me check if that makes sense.

If each aquarium has 14 fish, then the total number of fish in both aquariums is 28. The difference between the number of snails ($32 - 4 = 28$) is equal to twice the number of fish ($2 \times 14 = 28$). Yes, that checks out. So, the answer is 14 fish in each aquarium.

Wait, but let me make sure I didn't misinterpret the problem. The problem says "the difference between the number of snails in the two aquariums is twice the amount of fish in both aquariums." So, the difference in snails (28) is equal to twice the total number of fish ($2F$). So, $28 = 2F$, which means $F = 14$. That seems correct.

Let me just go through it one more time to ensure I didn't mix up any numbers.

Number of snails in first aquarium: 4 Number of snails in second aquarium: 32 Difference in snails: $32 - 4 = 28$ Let $F$ be the number of fish in each aquarium Total fish in both aquariums: $2F$ According to the problem: $28 = 2F$ Solving for $F$: $F = 14$

So, each aquarium has 14 fish. That seems correct. I don't see any mistakes in this reasoning. Just to double-check, if each aquarium has 14 fish, then the total number of fish is 28. The difference in the number of snails is 28, which is twice the total number of fish ($2 \times 14 = 28$). Yep, that matches the problem statement.

Therefore, the number of fish in each aquarium is 14.

**Final Answer** The number of fish in each aquarium is $\boxed{14}$.

### Case 3: Correct to Correct

### Question

Kylar went to the store to buy glasses for his new apartment. One glass costs $5, but every second glass costs only 60% of the price. Kylar wants to buy 16 glasses. How much does he need to pay for them?

1080
1081
1082
1083
1084
1085
1086
1087
1088
1089
1090
1091
1092
1093
1094
1095
1096
1097
1098
1099
1100
1101
1102
1103
1104
1105
1106
1107
1108
1109
1110
1111
1112
1113
1114
1115
1116
1117
1118
1119
1120
1121
1122
1123
1124
1125
1126
1127
1128
1129
1130
1131
1132
1133

**R1-1.5B's Answer**

**Note: The answer should be in a box.**

Alright, so Kylar wants to buy glasses for his new apartment, and he needs to figure out how much he has to pay for 16 glasses. Let me try to understand the problem step by step.

First off, the problem says that one glass costs $5. But here's the twist: every second glass costs only 60% of the price. Hmm, okay, so it's like every even-numbered glass is cheaper, right? So, if Kylar is buying 16 glasses, he'll have to buy 16 items, alternating between $5 and 60% of $5.

Let me break this down. Maybe I can figure out the cost of two glasses first, and then see how that pattern repeats for 16 glasses. That sounds manageable because 16 is an even number, so it should come out even.

So, let's think: the first glass is $5, and the second one is 60% off. What's 60% of $5? Hmm, 60% is the same as 0.6, right? So, 0.6 times 5. Let me calculate that. 0.6 * 5 = 3. So, the second glass is $3.

Okay, so every pair of glasses costs $5 + $3 = $8. That seems right. So, for every two glasses, Kylar spends $8.

Now, if he needs 16 glasses, how many pairs is that? Well, 16 divided by 2 is 8. So, there are 8 pairs of glasses. Each pair costs $8.

Therefore, to find the total cost, I can multiply the cost per pair by the number of pairs.

Total cost = 8 pairs * $8 per pair = $64.

Wait, is that all? Let me verify to make sure I didn't miss anything.

Each pair is two glasses: one at $5 and one at $3. So, two glasses for $8. 16 glasses mean 8 pairs, so 8*8 is indeed 64.

Let me think another way to double-check. Maybe I can calculate the total cost without grouping. So, for each glass, if it's an odd-numbered one, it's $5, and if it's even-numbered, it's $3.

So, let's list out the cost for each glass:

1st glass: $5
2nd glass: $3
3rd glass: $5
4th glass: $3
5th glass: $5
6th glass: $3
7th glass: $5
8th glass: $3
9th glass: $5
10th glass: $3
11th glass: $5
12th glass: $3
13th glass: $5
14th glass: $3
15th glass: $5
16th glass: $3

Now, let's count how many $5s and how many $3s there are.

Looking at the list, the odd-numbered glasses (1,3,5,7,9,11,13,15) are $5 each. That's 8 glasses at $5.

The even-numbered glasses (2,4,6,8,10,12,14,16) are $3 each. That's also 8 glasses at $3.

So, total cost is (8*$5) + (8*$3).

Calculating that:

8*$5 = $40

8*$3 = $24

Adding them together: $40 + $24 = $64.

Yep, same result as before. So, that seems consistent.

Is there another way to think about this? Maybe considering the average price per glass?

Wait, if every second glass is 60% of the price, that's effectively making every other glass 60% instead of 100%. Since 60% is less than 100%, the overall average price per glass is somewhere between $5 and $3.

Let me compute the average price.

Total cost is $64 for 16 glasses, so the average price is $64 / 16 = $4 per glass.

So, the average price is $4, which is exactly halfway between $3 and $5. Wait, hold on, no. $3 and $5 average to $4. So, actually, the average price per glass is $4.

But wait, that's interesting. So, if you have 16 glasses, each costing an average of $4, then total is 16*$4 = $64.

So, that's another way to see it.

Alternatively, I can think of each pair as a unit that costs $8, so 16/2 pairs = 8 units, 8*$8 = $64.

Either way, it seems the total is $64.

But just to be thorough, let me simulate it with smaller numbers.

Suppose Kylar buys 2 glasses. That should be $5 + $3 = $8.

If he buys 4 glasses: two pairs, so 2*$8 = $16.

Breaking it down: 1,2,3,4. So, $5, $3, $5, $3. Total is $5 + $3 + $5 + $3 = $16.

Similarly, 6 glasses: 3 pairs, so 3*$8 = $24.

1,2,3,4,5,6: $5, $3, $5, $3, $5, $3. Total is 3*$5 + 3*$3 = $15 + $9 = $24.

So, the pattern holds. So, for 16 glasses, it's 8 pairs, each worth $8, so 8*$8 = $64.

I think that makes sense.

Alternatively, I can think about the pricing structure as a sequence where every even term is 60% of the first term. So, it's a geometric sequence where each term is 0.6 times the previous term.

But wait, no, that might complicate things. Maybe it's better to model it as two separate arithmetic sequences: one for the odd-numbered glasses and one for the even-numbered glasses.

So, for odd-numbered glasses (1,3,5,...15): each is $5. There are 8 of them, so total is 8*$5 = $40.

For even-numbered glasses (2,4,6,...16): each is $3. There are 8 of them, so total is 8*$3 = $24.

Total cost is $40 + $24 = $64.

Yes, same answer again.

I wonder if someone might get confused and think that every glass is 60% of the first one, which would be $3, but that's not the case. It's every second glass, meaning every even-numbered glass is 60% of the price, not every glass. So, it's not a discount on all glasses, just on every second one.

So, in other words, the pricing alternates between $5 and $3 for each glass, with the even-numbered ones being $3. So, the pattern is consistent and straightforward.

Let me just do a quick check. If he buys 16 glasses, 8 of them are $5, 8 are $3. 8*$5 is 40, 8*$3 is 24. Total is 64.

Alternatively, if I tried to calculate the total by just multiplying 16 by the average price.

If the average price is $4, then 16*$4 is $64.

So, regardless of the method, the answer is $64.

I think I've considered multiple approaches and each time I come back to the same total. So, I'm pretty confident that the answer is $64.

**Final Answer**

Kylar needs to pay

boxed64 dollars for the glasses. </think>

Kylar wants to buy 16 glasses, where each glass costs $5, but every second glass costs 60% of the price.

First, we note that every pair of glasses consists of one at $5 and one at $3 (since 60% of $5 is $3). Each pair costs $8.

For 16 glasses, there are 8 pairs. Therefore, the total cost is calculated as follows:

$$8 \text{ pairs} \times 8 \text{ per pair} = \$64.$$

Thus, Kylar needs to pay $\boxed{64}$ dollars for the glasses.

**SelfBudgeter-1.5B's Answer**

Okay, so Kylar needs to buy 16 glasses for his new apartment, and the store is offering a deal:

one glass costs \$5, but every second glass costs only 60% of the price. Hmm, let me try to figure out how much he has to pay in total.

First, let me make sure I understand the problem correctly. The first glass is \$5, right? Then the second glass is 60% of \$5, the third glass is \$5 again, the fourth glass is 60% of \$5, and so on. So, it's like every even-numbered glass is 60% off, and the odd-numbered ones are full price. That makes sense. So, for 16 glasses, there will be 8 pairs of glasses, each pair consisting of one full price and one discounted price.

Let me break it down step by step. If each pair consists of one \$5 glass and one 60% discounted glass, then the cost for each pair would be:

$$\$5 + 0.6 \times \$5.$$

Since:

$$0.6 \times 5 = 3,$$

each pair costs:

$$5 + 3 = 8.$$

There are 16 glasses, so:

$$16 \div 2 = 8 \text{ pairs.}$$

Thus the total cost is:

$$8 \times 8 = \$64.$$

Wait, let me double-check. If each pair is \$8, then 8 pairs would be:

$$8 \times 8 = 64.$$

Let me verify again: Odd-numbered glasses: \$5 each Even-numbered glasses: \$3 each So:

$$8 \times 5 = 40, \quad 8 \times 3 = 24,$$
$$40 + 24 = 64.$$

Yes, that seems right.
Another approach: Without discount:

$$16 \times 5 = 80.$$

Savings per discounted glass:

$$5 - 3 = 2.$$

There are 8 discounted glasses, so:

$$8 \times 2 = 16.$$

Total:

$$80 - 16 = 64.$$

Yes, matches the earlier result.
So regardless of the method, the total cost is:

$$\$64.$$

**Final Answer**
Kylar needs to pay $\boxed{64}$ dollars for the 16 glasses.

# E ABLATION STUDY ON REWARD COMPONENTS

To understand the contribution of each reward component, we ablate SelfBudgeter by progressively adding format, correctness, preb, and budget-penalty terms. As shown in Table 5, learning format only fails to preserve problem-solving ability: accuracy on all datasets collapses (e.g., GSM8K 1.64%), despite producing shorter outputs. Adding correctness substantially restores accuracy, but compression becomes inconsistent, especially on harder datasets such as MATH500 and AIME2025, where reasoning lengths remain long (e.g., 5327 tokens on MATH500). Incorporating preb improves budget adherence but harms performance on easier tasks, leading to notable accuracy drops on GSM8K and AIME2025. Adding budget-penalty instead yields good compression on easy tasks but sacrifices performance on difficult ones (e.g., lower AIME2025 accuracy with long residual lengths). In contrast, SelfBudgeter, which integrates all reward components, achieves the best overall trade-off: high accuracy across datasets and strong, stable compression (e.g., 1231 tokens on GSM8K and 2327 tokens on MATH500). This demonstrates that the reward components are complementary—format ensures structure, correctness preserves reasoning quality, preb stabilizes predicted budgets, and the budget-penalty enforces compliance—and only their combined use yields robust reasoning-length control without degrading accuracy.

| Models | GSM8K | | MATH500 | | AIME2025 | |
|---|---|---|---|---|---|---|
| | Acc | Len | Acc | Len | Acc | Len |
| R1-1.5B | 73.09 | 2865.08 | 74.93 | 5327.12 | 22.22 | 14444.03 |
| Format only | 1.64 | 4866.40 | 1.47 | 5558.35 | 0.00 | 6448.66 |
| F+Correctness | 84.58 | 1258.45 | 76.27 | 3123.37 | 20.00 | 11095.39 |
| F+C+Preb | 82.89 | 2151.26 | 78.00 | 3109.43 | 15.56 | 4671.86 |
| F+C+BudgetPenalty | 84.26 | 1263.38 | 76.67 | 2983.28 | 21.11 | 7263.38 |
| SelfBudgeter-1.5B | 84.10 | 1231.79 | 78.47 | 2326.85 | 21.11 | 4288.10 |

Table 5: Ablation of Reward Components on Accuracy and Response Length

# F THE USE OF LARGE LANGUAGE MODELS

During the preparation of this paper, large language models (LLMs) were used solely as auxiliary tools for grammar checking, text polishing, and improving clarity of exposition. No experimental design, data analysis, or substantive research conclusions were generated by LLMs. All methodological and experimental contributions are original and conducted entirely by the authors.

