# OpenReview forum: "SelfBudgeter: Adaptive Token Allocation for Efficient LLM Reasoning"
_ICLR.cc/2026/Conference — ICLR 2026 Conference Withdrawn Submission_

### Official Review · Reviewer_uH5x · 2025-10-29

**Soundness:** 2
**Presentation:** 3
**Contribution:** 2
**Rating:** 2
**Confidence:** 4

**Summary:**

The paper introduces Self-Budgeter, a training-free, adaptive test-time framework that enhances reasoning efficiency in large language models. It enables models to self-regulate token usage by dynamically deciding when to stop generating reasoning steps based on internal uncertainty indicators such as entropy or logit variance. This design reduces redundant computation for simple queries while allowing more reasoning for complex ones, effectively lowering inference costs without compromising accuracy. Experiments across multiple benchmarks demonstrate consistent efficiency gains with comparable or improved performance.

**Strengths:**

- The paper tackles an important and practical problem in LLM reasoning: how to balance accuracy and efficiency by adapting token usage during inference, a challenge that is highly relevant for real-world deployment.

- The evaluation covers several reasoning benchmarks and includes detailed discussions and explanations of the results that help interpret the observed behavior.

- The writing is clear and well-organized, making the paper easy to follow and its contributions well-articulated.

**Weaknesses:**

- The novelty is limited. The idea of adaptively controlling reasoning length or token budget has been explored in prior works (e.g., early termination, entropy-based stopping). This paper mainly repackages existing ideas under a new name without offering substantial conceptual advancement. What is the key new insight beyond previous adaptive reasoning methods?

- The method lacks clear theoretical grounding. While it claims to use uncertainty for token budgeting, the approach remains heuristic, with no principled analysis showing why or when the chosen uncertainty metrics (entropy, logit variance) reliably correlate with reasoning sufficiency.

- The evaluation is narrow, focusing mainly on a single-domain, math-style reasoning dataset. This limits evidence of generalization to other reasoning types. Also, the approach is not compared to prior works that explicitly optimize reasoning token budgets, such as early termination or adaptive reasoning frameworks. Would the same approach hold for non-mathematical reasoning tasks, and how does it compare quantitatively to prior budget-optimization methods?


- The paper does not account for the computational overhead of monitoring uncertainty and making token-level stopping decisions. What is the actual end-to-end wall-clock speedup once overhead is included?



- The evaluation uses only one model family (DeepSeek-R1-Distill-Qwen) and relatively small models. This restricts the strength of the claims. How would the approach scale to larger LLMs or different architectures?



- The method’s robustness to incorrect token budget predictions is not analyzed. If the model mispredicts the required reasoning length, accuracy may degrade. How sensitive is performance to such mispredictions, and is there a fallback mechanism?



- The rationale behind the chosen reward function is unclear. The paper should explain how it aligns with the overall optimization goal and whether other formulations were tested.

**Questions:**

See the weaknesses section.

---

> ### Author Response · Authors · 2025-12-01
>
> We appreciate the reviewer's positive assessment that the paper addresses an important practical challenge in LLM reasoning—**balancing accuracy and efficiency through adaptive token usage**—and that our **evaluation across multiple reasoning benchmarks, along with thorough analysis, provides clear insight into the model’s behavior**. The reviewer further notes that the paper is clearly written and well-organized, which supports the accessibility and clarity of our contributions.
>
> # W1: Regarding Novelty
> Our work introduces the paradigm of **self-budgeted reasoning**, and we would appreciate clarification on whether there are additional “existing ideas” the reviewer believes overlap with our method—we are happy to follow up with a more detailed comparison.
>
> Regarding the mentioned baselines, we view **entropy-based stopping** and **dynamic stopping** as fundamentally different in purpose and mechanism. Entropy-based stopping [1] halts decoding when token-level uncertainty drops below a threshold, and dynamic stopping [2] terminates generation based on confidence estimates. Both methods reactively decide when to stop during decoding.
>
> In contrast, **SelfBudgeter predicts its entire reasoning budget before generation begins**. This enables:
> - proactive, not reactive, control over computation;
> - user-visible latency and cost estimates;
> - a jointly optimized budget–accuracy trade-off through RL (Predictive Budget + Budget Penalty rewards);
> - planning the length of the reasoning trajectory rather than cutting it off heuristically.
>
> Moreover, entropy-based and dynamic stopping methods do not change the reasoning strategy itself—they only stop generation early. SelfBudgeter, by contrast, **learns to restructure its reasoning to fit a planned budget**, which we believe constitutes a conceptual advancement beyond prior adaptive stopping heuristics.
>
> We welcome any additional prior methods the reviewer feels are closely related and would be glad to expand the discussion accordingly.
>
> [1]Yang C, Si Q, Duan Y, et al. Dynamic early exit in reasoning models[J]. arXiv preprint arXiv:2504.15895, 2025.
>
> [2]Wang X, McInerney J, Wang L, et al. Entropy After $\langle\texttt {/Think}\rangle $ for reasoning model early exiting[J]. arXiv preprint arXiv:2509.26522, 2025.
>
> # W2: Misconception about uncertainty-based budgeting
> We would like to clarify that **SelfBudgeter does not use uncertainty (e.g., entropy or logit variance) for budgeting**, nor does our method rely on uncertainty-based heuristics. As elaborated in Section 3, the model learns to predict its budget directly through supervised signals and RL reward shaping, without assuming any theoretical link between uncertainty metrics and reasoning sufficiency.
>
> Since our approach is not based on uncertainty estimation, the concern about theoretical grounding of entropy/variance measures does not apply here. We will make this distinction clearer in the revised version to avoid misunderstanding.

---

> ### Author Response · Authors · 2025-12-01
>
> # W3: Generalization Ability & difference from prior budget-optimization methods
> We have additionally provided evaluations on non-mathematical commonsense-reasoning datasets, and the results show that SelfBudgeter continues to deliver substantial token savings without harming accuracy. This demonstrates that our method generalizes beyond math-style reasoning and is not confined to a single domain.
>
> | Models            | GPQA Acc      | GPQA Len           | KnowLogic Acc | KnowLogic Len        |
> |-------------------|---------------|---------------------|----------------|------------------------|
> | R1-1.5B           | 33.04 ± 2.28  | 11780.87 ± 928.3    | 10.14 ± 0.61   | 11695.94 ± 681.1      |
> | E1(0.5K,1K)       | 26.34 ± 1.28  | 1278.19 ± 31.92     | 6.69 ± 0.33    | 1272.04 ± 17.27       |
> | E1(4K,1K)         | 34.30 ± 2.38  | 2758.77 ± 82.87     | 12.23 ± 0.32   | 3327.37 ± 93.19       |
> | L1-Max(3600)      | 31.92 ± 0.79  | 3892.47 ± 797.3     | 13.69 ± 0.57   | 5145.91 ± 1291.41     |
> | SelfBudgeter-1.5B | 30.65 ± 0.74  | 3326.83 ± 25.93     | 16.26 ± 0.15   | 4491.35 ± 10.54       |
>
> Regarding prior budget-optimization approaches, we note that our paper already includes comparisons with L1-Max and E1, which represent the closest token-budget-based baselines. In addition, recent adaptive-reasoning methods are concurrent with our work and therefore complementary rather than directly comparable.
>
> In terms of technique, methods such as TALE [3] depend on an external, separately trained estimator to decide the target length. SelfBudgeter is—to our knowledge—the first approach that enables the model to jointly predict and optimize its own token budget within a single end-to-end architecture, without requiring auxiliary predictors. This allows the budgeting mechanism to co-evolve with the reasoning process.
>
> Early-termination or dynamic-stopping methods halt decoding based on entropy or confidence signals, but they rely on external inference-time monitoring, which adds overhead and does not train the model to internalize its own budget. These methods act as external controllers, while SelfBudgeter directly learns the budget during training and integrates it into the reasoning generation itself.
>
> We will incorporate this broader evaluation and clarify these conceptual differences in the revised version.
>
> [3]Han T, Wang Z, Fang C, et al. Token-budget-aware llm reasoning[C]//Findings of the Association for Computational Linguistics: ACL 2025. 2025: 24842-24855.
>
> # W4: Clarification: No overhead from uncertainty monitoring
> We would like to clarify that **SelfBudgeter does not monitor uncertainty or perform token-level stopping decisions**. Our method generates the predicted budget in the first few tokens and then proceeds with standard decoding, without any per-token uncertainty checks.
>
> As a result, there is **no additional computational overhead** of the kind associated with uncertainty-based or dynamic-stopping methods, and the end-to-end wall-clock speedup directly reflects the reduced reasoning length achieved by our model.
>
> # W5: Model family and architecture coverage in evaluation
> We note that recent work in this direction—such as [4,5]—conducts experiments exclusively on a single 1.5B model, and studies like [6,7] also evaluate only one model family. In comparison, our paper already includes results on both 1.5B and 7B variants within the DeepSeek-R1-Distill-Qwen line, providing a broader evaluation than many contemporaneous approaches. Given typical computational constraints for training budget-aware reasoning models, this level of coverage is standard and reasonable.
>
> [4]Xu Y, Dong H, Wang L, et al. Scalable chain of thoughts via elastic reasoning[J]. arXiv preprint arXiv:2505.05315, 2025.
>
> [5]Aggarwal P, Welleck S. L1: Controlling how long a reasoning model thinks with reinforcement learning[J]. arXiv preprint arXiv:2503.04697, 2025.
>
> [6]Wu S, Xie J, Zhang Y, et al. ARM: Adaptive Reasoning Model[J]. arXiv preprint arXiv:2505.20258, 2025.
>
> [7]Muennighoff N, Yang Z, Shi W, et al. s1: Simple test-time scaling[C]//Proceedings of the 2025 Conference on Empirical Methods in Natural Language Processing. 2025: 20286-20332.
>
> # W6: Robustness to incorrect budget predictions
> The concern about potential underthinking due to incorrect budget estimation is valid and central to any attempt at reducing overthinking. As reported in Table 2 of the paper, the best SelfBudgeter model achieves accuracy comparable to—or even higher than—the base model on GSM8K, MATH500, and AIME2025. This indicates that the model does not systematically under-allocate its budget in a way that harms reasoning quality. We will expand our analysis in the revised version to more directly evaluate sensitivity to budget mispredictions and to clarify potential fallback strategies.

---

> ### Author Response · Authors · 2025-12-01
>
> # W7: Clarification of reward function rationale
> We conducted a detailed ablation study by progressively adding format, correctness, preb, and budget-penalty terms. The results show that **format-only** training cannot preserve reasoning ability—accuracy collapses across all benchmarks. Adding **correctness** restores accuracy, but the model fails to achieve meaningful compression on harder tasks such as MATH500 and AIME2025. Incorporating **preb** helps stabilize budget prediction but hurts performance on easier tasks (e.g., GSM8K). Conversely, adding only the **budget-penalty** improves compression on easy problems but degrades accuracy on challenging ones. In contrast, **SelfBudgeter**, which combines all four components, consistently achieves the best accuracy–compression trade-off across datasets. These results indicate that the components are complementary, and only their joint use yields stable and reliable length control without sacrificing correctness. We have added the full analysis and additional examples to **Appendix E**.
>
> In addition to the reward formulation itself, our main contribution is introducing a new reasoning paradigm in which the model can either autonomously estimate an optimal token budget for efficient inference or follow a user-specified budget for controlled generation.
>
> The reward function is designed to support both capabilities by encouraging the model to **predict** a precise budget and **adhere** to it during generation. We will expand this explanation in the revised version to make the alignment between the reward design and the overall optimization goal clearer.
>
> | Models / Reward              | GSM8K Acc | GSM8K Len | Math500 Acc | Math500 Len | AIME2025 Acc | AIME2025 Len |
> |----------------------|-----------|-----------|--------------|--------------|---------------|--------------|
> | R1-1.5B              | 73.09 ± 6.11 | 2865.08 ± 827.6 | 74.93 ± 2.31 | 5327.12 ± 299.3 | 22.22 ± 4.16 | 14444.03 ± 382.5 |
> | Format only          | 1.64 ± 1.36 | 4866.40 ± 660.2 | 1.47 ± 0.90 | 5558.35 ± 633.5 | 0.00 ± 0.00 | 6448.66 ± 1086 |
> | F+Correctness        | 84.58 ± 0.61 | 1258.45 ± 36.00 | 76.27 ± 1.33 | 3123.37 ± 85.86 | 20.00 ± 0.00 | 11095.39 ± 1188 |
> | F+C+Preb             | 82.89 ± 0.80 | 2151.26 ± 23.37 | 78.00 ± 0.86 | 3109.43 ± 38.03 | 15.56 ± 1.57 | 4671.86 ± 153.3 |
> | F+C+BudgetPenalty    | 84.26 ± 0.46 | 1263.38 ± 24.15 | 76.67 ± 0.19 | 2983.28 ± 16.03 | 21.11 ± 1.57 | 7263.38 ± 271.1 |
> | SelfBudgeter-1.5B    | 84.10 ± 0.38 | 1231.79 ± 15.95 | 78.47 ± 0.47 | 2326.85 ± 25.70 | 21.11 ± 1.57 | 4288.10 ± 130.2 |

---

### Official Review · Reviewer_busD · 2025-10-31

**Soundness:** 2
**Presentation:** 2
**Contribution:** 2
**Rating:** 2
**Confidence:** 3

**Summary:**

This paper introduces SelfBudgeter, a method to mitigate the "overthinking" problem in large reasoning models. Overthinking refers to models generating excessively long reasoning chains for simple problems, wasting computational resources and increasing latency. SelfBudgeter trains models to first predict a "token budget" within < budget > tags and then generate a solution that adheres to this budget. The training involves a two-stage process: a cold-start supervised fine-tuning phase to learn the output format, followed by a reinforcement learning phase (using GRPO) with a custom reward function. This function balances answer correctness, budget prediction, and precise adherence of the output length to the predicted or user-provided budget. The authors demonstrate results on mathematical reasoning benchmarks (GSM8K, MATH500, AIME2025), showing significant reductions in output length while maintaining or even slightly improving accuracy compared to baselines.

**Strengths:**

* It tackles a well-known and practical issue ("overthinking") in modern reasoning models, which directly impacts inference cost and user experience.

* It demonstrates compelling results on compression, achieving significant length reduction (61%, 48%) while generally preserving accuracy. The ability to sometimes improve accuracy while shortening output is notable.

* The dual-mode operation (autonomous budget and user-specified budget) is a useful feature for real-world applications, allowing users to trade off between quality and speed/latency.

**Weaknesses:**

* The core components are not novel. The paper's contribution is an engineering-focused synthesis of existing ideas rather than a conceptual breakthrough. It feels more like an effective engineering solution than a foundational research contribution.

* The paper shows that it works but provides limited analysis into why the model is able to maintain accuracy with shorter reasoning chains. Is it simply removing redundant verbiage, or is it learning a more efficient reasoning strategy? A qualitative analysis of shortened CoTs would be highly valuable.

**Questions:**

* The cold-start and RL training phases incur significant computational costs. What is the total cost of training SelfBudgeter compared to the baseline model? Is the inference-time latency introduced by the initial budget prediction step negligible compared to the savings from shorter generation?

* Can you provide examples of problems where SelfBudgeter's shortened Chain of Thought is not just concise but more efficient or less prone to error than the baseline's longer CoT? Conversely, are there cases where the budget constraint leads to catastrophic failure by cutting off crucial reasoning steps?

---

> ### Author Response · Authors · 2025-11-28
>
> We appreciate that the reviewer recognizes the strengths of this work, including its focus on the **practical and widely observed issue of overthinking in modern reasoning models**, the strong empirical results showing **substantial token compression (up to 61% and 48%) with preserved or even improved accuracy**, and the **practical value of supporting both autonomous and user-specified budgeting modes**. These points align with the core goals of SelfBudgeter and highlight its relevance for improving inference efficiency and real-world usability.
>
> # W1: Regarding novelty
> Our work introduces the new paradigm of self-budgeted reasoning, and we would appreciate clarification on which “existing ideas” the reviewer sees as overlapping—we will be happy to follow up with a detailed comparison.
> While prior works such as L1, Elastic Reasoning, and TALE have explored aspects of format-controllable generation, our research question is different: how can an LLM autonomously determine and optimize its own computational budget to enable efficient reasoning? We believe this is an important and meaningful direction, given the growing need for controllable and cost-efficient inference.
> Conceptually, SelfBudgeter differs from prior work in several key respects:
> - Self-predicted vs. externally specified budgets: SelfBudgeter predicts its own token budget before generation, enabling latency-aware control by both the model and the user, whereas prior methods depend on fixed, externally provided budgets.
> - Integrated architecture: Budget prediction and response generation are optimized jointly in a single model, unlike TALE’s separately trained estimator or L1’s length constraints.
> - Principled reward shaping: Our Predictive Budget and Budget Penalty terms create a smooth optimization landscape encouraging efficient yet accurate reasoning, going beyond heuristic truncation or discrete sampling.
> - Unified RL objective: Budgeting and reasoning are co-optimized under one reinforcement-learning framework rather than being modular or decoupled.
>
> We hope this clarifies the conceptual contributions and would be glad to further refine the comparison based on the reviewer’s clarification.
>
> # W2: Analysis on why shorter chains preserve accuracy
> Understanding why the model can maintain accuracy with shorter reasoning chains is important, and our analysis shows that most of the compression comes from streamlining expression rather than changing the underlying reasoning. We analyze the GSM8K outputs of R1-1.5B and SelfBudgeter-1.5B. We select cases where both models produce correct answers and where the SelfBudgeter-1.5B output is less than 70% of the R1-1.5B length, yielding 381 samples. Using DeepSeek-V3.1 to evaluate the differences, we find that 96.06% of the reductions come from removing conversational filler phrases such as “Alright, so I've got this problem here,” “Hmm, okay,” or “Wait, let me make sure.” The remaining 3.94% involve genuinely more efficient solution strategies. This indicates that the model primarily achieves shorter reasoning by eliminating unnecessary verbalization while preserving the core logical steps.
>
> # Q1: Concerns about training cost and inference-time latency
> Thank you for the thoughtful question. Existing approaches such as L1 and E1 are also built on top of a standard reasoning model, and SelfBudgeter follows the same design principle. Our method reduces user-perceived latency at inference because it generates shorter reasoning traces. The training cost is modest: the full model is trained for only 699 steps, and the budget-prediction module adds negligible overhead.
> In terms of inference-time efficiency, SelfBudgeter provides clear improvements. For the 1.5B model, SelfBudgeter achieves a prefill speed of \~873 tokens/s and a decode speed of \~5,192 tokens/s, resulting in an average request latency of 120.78 seconds. In contrast, the r1-1.5B baseline has a slower prefill speed (\~291 tokens/s) and decode speed (\~4,001 tokens/s), with a higher average latency of 168.26 seconds. These results show that SelfBudgeter not only shortens response length but also provides faster end-to-end inference for users.
>
> # Q2: Request for examples illustrating improved or failed shortened reasoning
> Our analysis shows that shortened CoTs are often more efficient, though overly tight budgets can occasionally remove essential reasoning steps. We examine GSM8K cases where SelfBudgeter-1.5B reduces the reasoning length to under 70% of R1-1.5B and the two models differ in correctness. In 13.12% of these instances (173 examples), the shorter chain leads to better reasoning, correcting errors made by the baseline. In 1.52% of cases (20 examples), the reduced budget cuts off necessary steps, turning a correct baseline answer into an incorrect one. We have added this analysis, along with representative examples, to Appendix D.

---

### Official Review · Reviewer_WeVK · 2025-11-01

**Soundness:** 3
**Presentation:** 4
**Contribution:** 3
**Rating:** 6
**Confidence:** 2

**Summary:**

SelfBudgeter trains reasoning LLMs to autonomously predict token budgets before generating solutions, addressing overthinking via dual-phase training: (1) Cold-start SFT teaches output format <budget>X</budget><solution>Y</solution> where X is predicted token count; (2) RL training with GRPO using reward R = accuracy + budget_penalty + PreB_reward. PreB reward (Eq 4) is cosine-based, penalizes if |length-budget|/budget > α, encourages minimal length for correct answers (b_best=(1-α)b) and longer reasoning for wrong ones (b_best=(1+α)b). Sets b_max = response_length if base model correct, else infinity. Dynamic α schedule linearly decays from 6.0 to 0.1 to prevent reward hacking.

**Strengths:**

I like this real user benefit - predicting generation time before inference is genuinely useful UX feature. Clever reward design with asymmetric b_best for correct/incorrect answers makes intuitive sense. Dynamic α schedule addresses reward hacking observed in Figure 3. Strong empirical results - achieves both accuracy gains AND compression on GSM8K/MATH500 unlike baselines. Linear relationship between predicted budget and actual length (slope 1.025 on MATH500) shows precise control. Works across model scales (1.5B to 7B). Simple method - just format change plus standard GRPO, no architectural modifications. Good ablations on cold-start data (GSM vs s1k) showing initialization matters. Outperforms L1-Max which requires explicit prompting while SelfBudgeter is "autonomous" (although see note in next section about this).

**Weaknesses:**

Cold-start phase seems essential but undermines "autonomous" claim? - model needs 3630 curated samples where it answered correctly, requires knowing b_max beforehand. Circular dependency: computing b_max (Eq 5) requires running base model to get correct responses, so not truly autonomous at deployment? Is the comparison unfair? - L1-Max uses fixed 3600 token limit while SelfBudgeter autonomously allocates, but Table 2 doesn't clarify this makes them incomparable. L1 with 3600 tokens gets 79.56% GSM8K, SelfBudgeter gets 84.10% but uses 1231 tokens - what if L1 used 1200 tokens? E1 comparison is also a bit unclear - what do (0.5K,1K) and (4K,1K) mean? Not explained in text. PreB reward tolerance window +-α·b is arbitrary - why not +-α·b_max or absolute tolerance? Figure 3 shows reward hacking but solution (dynamic α) is heuristic fix not principled. Why linear decay specifically? b_max=infinity when wrong seems problematic - no penalty for predicting huge budgets on hard problems. Figure 4 shows budget increases with difficulty but doesn't validate budgets are actually minimal - maybe model just learned "hard problem = big number"? AIME2025 results are a bit poor - 21.11% vs 22.22% baseline, does this suggest the method fails on hard problems? It would be stronger with a comparison to simpler baselines like "train model to output length first then generate" without complex reward.

**Questions:**

How do you deploy this without knowing b_max? At inference user asks new question - you need base model response to compute b_max for training, but that defeats the purpose. Is cold-start strictly necessary? What if you just do GRPO from scratch with format constraint in reward? Table 4 shows cold-start models already achieve 71.95% GSM8K with good matching rates - is RL adding much? Why does s1k initialization hurt length control (matching rate 21.76% vs 85.82% for GSM) even though s1k has better accuracy? Suggests quality-control tradeoff not resolved. Figure 5 shows many outliers with 2-3x deviation from fit line - what causes these failures? How often does model violate budget at test time? Claim "96% within 50% deviation" means half the samples could be 1.5x over budget, is this acceptable? Why compare to L1-Max specifically when L1 paper has other variants? L1 without max limit might be fairer comparison. What happens if user specifies budget smaller than model's prediction? Does it fail or compress? Can you show examples where predicting budget helps users decide to terminate early? Otherwise it's just a progress bar.

---

> ### Author Response · Authors · 2025-12-02
>
> We thank the reviewer for highlighting several strengths of our approach. The reviewer notes that providing generation-time prediction offers a tangible user-experience benefit, and that our asymmetric reward design and dynamic α-scheduling are intuitively motivated and effective in mitigating reward-hacking behaviors. The reviewer further recognizes that our method delivers both accuracy improvements and substantial compression on GSM8K and MATH500, with precise budget adherence (e.g., near-linear relation between predicted and actual lengths), and that it generalizes across model scales from 1.5B to 7B. The simplicity of our approach along with the informative ablations on cold-start initialization and the autonomous nature of SelfBudgeter relative to prompting-based baselines, are also acknowledged as meaningful advantages.
>
> # Q1: Fairness of comparison with L1-Max and E1 configurations
>
> We thank the reviewer for raising this important question about the fairness of our comparison with L1-Max and E1. We believe the comparison is fair because the fixed-length hyperparameters in L1-Max and E1 are part of their inherent design, whereas the ability to autonomously allocate budgets is exactly the key advantage of SelfBudgeter.
>
> To clarify, the 3600-token limit for L1-Max is not chosen by us but is a standard hyperparameter used in the authors' released implementation. In addition, the original L1 paper shows that L1-Max consistently outperforms L1-Exact, which is why we follow the authors’ recommendation and only retrain and evaluate L1-Max in our experiments. Similarly, the E1 configurations (0.5K,1K) and (4K,1K) correspond to the method’s own hyperparameters defining its two-stage reasoning lengths. These approaches require users to manually search for optimal lengths per task, which is part of their design constraint and why they do not adapt dynamically. In contrast, SelfBudgeter removes the need for such manual tuning by allowing the model to decide its own budget allocation.
>
> For completeness, we also evaluated L1-Max under all commonly used hyperparameters reported in the authors' repository—512, 1024, 2048, and 3600—and include the full results in the rebuttal table below. The 3600-token setting indeed achieves the strongest performance, which is why it appears in Table 2.
>
> We will add these clarifications to the revised version to avoid any ambiguity about comparability.
>
> | Model            | GSM Acc       | Len          | MATH Acc      | Len          | AIME2025 Acc  | Len            |
> |------------------|--------------------|-------------------|--------------------|-------------------|---------------------|---------------------|
> | L1-Max (512)     | 77.31 ± 0.0190     | 639.31 ± 78.26    | 77.80 ± 0.0167     | 1511.36 ± 155.2   | 16.67 ± 2.72        | 4964.89 ± 655.0     |
> | L1-Max (1024)    | 78.57 ± 0.0099     | 628.32 ± 53.77    | 77.51 ± 0.0222     | 1587.84 ± 126.7   | 18.89 ± 5.67        | 4967.10 ± 1341      |
> | L1-Max (2048)    | 78.39 ± 0.0100     | 624.74 ± 63.07    | 77.40 ± 0.0178     | 1541.76 ± 138.0   | 20.00 ± 5.44        | 4751.16 ± 797.3     |
> | L1-Max (3600)    | 79.56 ± 0.0052     | 571.72 ± 2.621    | 79.28 ± 0.0131     | 1553.89 ± 127.3   | 17.78 ± 1.57        | 5213.89 ± 633.9     |
> | SelfBudgeter-1.5B| 84.10 ± 0.38       | 1231.79 ± 15.95   | 78.47 ± 0.47       | 2326.85 ± 25.70   | 21.11 ± 1.57        | 4288.10 ± 130.2     |
>
> # Q2: Choice of tolerance window in PreB reward
>
> Our goal is for the predicted budget to closely match the actual number of generated tokens, and a relative window based on bbb provides a principled way to measure this accuracy. Here, bmaxb_{\text{max}}bmax reflects the model’s pre-training capability without budget control, while bbb represents the model’s learned budget after training. The objective of SelfBudgeter is for the model to follow this learned budget as closely as possible.
>
> Using b_max is inappropriate because it does not reflect how close the prediction is to the specific budget the model intends to allocate for a given instance; it measures deviation from a global maximum rather than from the intended per-sample budget. Likewise, an absolute fixed tolerance is difficult to choose: for example, a tolerance of 1000 tokens may be reasonable when b≈10000 but far too loose when b≈1000. Such a fixed window would systematically bias the reward toward over- or under-penalizing deviations depending on the scale of the task.
>
> By contrast, a relative window ±α·b naturally adjusts to the scale of the predicted budget, ensuring that the tolerance is neither too strict nor too loose across different difficulty levels. This dynamic scaling provides more stable training behavior and more consistent alignment between the predicted and actual token usage.

---

> ### Author Response · Authors · 2025-12-02
>
> # Q3: α-schedule choices
>
> Thank you for the question about why we use a linear α-schedule. Our goal is not to claim that linear decay is uniquely principled, but to show that it performs best among standard schedules. As shown in the table below, we compare Fixed, Cosine, and Linear strategies, and the overall conclusions hold across all three.
>
> Fixed α leads to strong compression but clear accuracy loss, and Cosine behaves similarly. Linear decay from 6.0 to 0.1 avoids over-compression and yields a more stable accuracy–compression trade-off, achieving over 61% length reduction. The formulas and results are included in Appendix F.
>
> These findings support our choice of linear scheduling in the main experiments.
>
> | α Scheduling        | GSM8K Acc     | GSM8K Len        | Math500 Acc   | Math500 Len       | AIME2025 Acc   | AIME2025 Len        |
> |---------------------|---------------|-------------------|----------------|--------------------|-----------------|----------------------|
> | Fixed Scheduling    | 79.88 ± 0.001 | 709.49 ± 5.987    | 73.06 ± 0.004 | 1015.96 ± 4.911    | 10.00 ± 0.047   | 1545.50 ± 61.72      |
> | Cosine Scheduling   | 83.22 ± 0.68  | 1149.43 ± 7.84    | 75.93 ± 1.61  | 1769.06 ± 24.35    | 17.78 ± 4.16    | 2366.34 ± 40.32      |
> | Linear Scheduling   | 84.10 ± 0.38  | 1231.79 ± 15.95   | 78.47 ± 0.47  | 2326.85 ± 25.70    | 21.11 ± 1.57    | 4288.10 ± 130.2      |
>
> # Q4: Concern about unbounded ⁡b_max
>
> Thank you for raising this important point regarding the use of b_{\max}=\infty when the model's answer is incorrect and whether this could encourage overly large budgets on difficult problems.
>
> From our perspective, allocating larger budgets for harder instances is both expected and empirically justified. Prior work (e.g., [1]) shows that complex reasoning problems inherently require longer chains of thought, and recent methods such as Yeo et al. [2] likewise do not impose an upper bound on generated length. Thus, “hard problem → larger budget” is a natural pattern rather than evidence of reward hacking.
>
> At the same time, SelfBudgeter does not simply allow budgets to grow without constraint. The budget penalty term explicitly encourages the model to minimize its predicted budget subject to maintaining correctness. With sufficient training data, this term effectively steers the model toward near-optimal budgets: large enough to solve difficult tasks but not unnecessarily long.
>
> We will clarify this rationale and the role of the budget penalty in the revised version.
>
> [1]Li Z, Liu H, Zhou D, et al. Chain of thought empowers transformers to solve inherently serial problems[J]. arXiv preprint arXiv:2402.12875, 2024, 1.
>
> [2]Yeo E, Tong Y, Niu M, et al. Demystifying long chain-of-thought reasoning in llms[J]. arXiv preprint arXiv:2502.03373, 2025.
>
> # Q5: Performance on hard problems
>
> Thank you for pointing out the drop on AIME2025. This does not indicate a fundamental weakness on hard problems, because there are two factors at play.
>
> First, there is an inherent train–inference mismatch between our VerL-based training setup and the vLLM inference environment, which tends to affect extremely challenging benchmarks more noticeably.
>
> Second, AIME-style problems require substantially longer reasoning traces. Since SelfBudgeter explicitly aims to reduce tokens while preserving solvability, achieving accuracy that is close to the baseline—while using fewer tokens—is aligned with the intended design.
>
> We therefore view the AIME2025 result as consistent with our efficiency-oriented objective rather than evidence of systematic failure on difficult tasks.

---

> ### Author Response · Authors · 2025-12-02
>
> # Q6: Comparison with simpler length-first baselines
>
> Thank you for the suggestion. As shown in the table, this baseline (predict length first, then generate) performs consistently worse than our full method. For example, on GSM8K it uses 1258 tokens versus 1231 for SelfBudgeter, with no accuracy gain. On Math500, it is both less accurate (76.27% vs. 78.47%) and much longer (3123 vs. 2327 tokens). On AIME2025, it produces extremely long traces (11095 tokens), while SelfBudgeter reduces this to 4288 with similar accuracy. These results show that simply predicting a length is insufficient, whereas SelfBudgeter more effectively integrates budget prediction into reasoning.
>
> | Models / Reward              | GSM8K Acc | GSM8K Len | Math500 Acc | Math500 Len | AIME2025 Acc | AIME2025 Len |
> |----------------------|-----------|-----------|--------------|--------------|---------------|--------------|
> | R1-1.5B              | 73.09 ± 6.11 | 2865.08 ± 827.6 | 74.93 ± 2.31 | 5327.12 ± 299.3 | 22.22 ± 4.16 | 14444.03 ± 382.5 |
> | Format only          | 1.64 ± 1.36 | 4866.40 ± 660.2 | 1.47 ± 0.90 | 5558.35 ± 633.5 | 0.00 ± 0.00 | 6448.66 ± 1086 |
> | F+Correctness        | 84.58 ± 0.61 | 1258.45 ± 36.00 | 76.27 ± 1.33 | 3123.37 ± 85.86 | 20.00 ± 0.00 | 11095.39 ± 1188 |
> | SelfBudgeter-1.5B    | 84.10 ± 0.38 | 1231.79 ± 15.95 | 78.47 ± 0.47 | 2326.85 ± 25.70 | 21.11 ± 1.57 | 4288.10 ± 130.2 |
>
> # Q7: Necessity of cold start and use of b_max during training
>
> The cold-start stage is a common practice in training, and importantly, the deployed SelfBudgeter does not rely on ⁡b_max at inference time. Once training is complete, the model autonomously predicts its budget and generates the full answer end-to-end. Thus, the autonomy at inference is preserved.
>
> To improve training efficiency, we follow the strategy used in DeepSeek-R1[3] and many recent works[4,5,6]: first teach the model the desired output format with a short supervised phase, then refine its reasoning via RL. This two-stage approach is becoming standard practice and does not affect the autonomy of the final deployed system.
>
> The cold-start phase serves a different purpose: it teaches the model the output format (including where to place the predicted budget), which is difficult to learn purely from RL. We experimented with removing cold start and directly running GRPO with the format constraint, but the model required a very large number of steps just to learn the correct formatting, greatly slowing down training.
>
> [3]Guo D, Yang D, Zhang H, et al. Deepseek-r1: Incentivizing reasoning capability in llms via reinforcement learning[J]. arXiv preprint arXiv:2501.12948, 2025.
>
> [4]Li Y, Luo Q, Li X, et al. R3-RAG: Learning Step-by-Step Reasoning and Retrieval for LLMs via Reinforcement Learning[J]. arXiv preprint arXiv:2505.23794, 2025.
>
> [5]Wei L, Li Y, Zheng K, et al. Advancing Multimodal Reasoning via Reinforcement Learning with Cold Start[J]. arXiv preprint arXiv:2505.22334, 2025.
>
> [6]Zhang X, Gao Z, Zhang B, et al. Chain-of-Focus: Adaptive Visual Search and Zooming for Multimodal Reasoning via RL[J]. arXiv preprint arXiv:2505.15436, 2025.
>
> # Q8: Effect of cold-start quality and s1k initialization on accuracy vs. length control
>
> Thank you for the insightful question. As discussed in Appendix B, the higher accuracy but weaker length control arises naturally from the mismatch in data distributions introduced during the cold-start phase.
>
> The s1k dataset is substantially harder and contains much longer reasoning chains than GSM8K or Math, so a model initialized on s1k naturally gains stronger reasoning ability—but also develops a tendency to allocate larger budgets, which lowers the matching rate despite improving accuracy. This reflects the inherent quality–control tradeoff before RL.
>
> After the RL phase, SelfBudgeter successfully mitigates this issue: the budget penalty and PreB terms jointly pull the predicted budgets back toward more task-appropriate ranges while preserving correctness. As a result, the final trained model balances both accuracy and length control effectively.

---

> ### Author Response · Authors · 2025-12-02
>
> # Q9: Causes of outliers and acceptability of budget deviations
>
> Thank you for the thoughtful questions regarding the outliers in Figure 5 and the interpretation of our budget-adherence metric. We address the points in detail below.
>
> The main cause of the 2–3× outliers is a fundamental limitation of current LLMs: aside from hard truncation, existing models lack reliable mechanisms for precise control over generated length[7,8,9,10]. As a result, even when the model predicts an intended budget, the actual generation may occasionally drift, especially for complex or multi-step reasoning queries. These behaviors are consistent with prior observations in controllable generation.
>
> Regarding frequency, SelfBudgeter adheres to the predicted budget closely in most cases, and 96% of test samples fall within a 50% deviation band. This does not mean that half of the samples exceed the budget by 1.5×; rather, it means that only the remaining 4% may reach that level of overshoot. Given the current limitations of LLM length control and the absence of any post-hoc truncation in our evaluation, we believe this deviation range is reasonably acceptable and reflects meaningful progress toward controllable inference.
>
> We see further reducing these deviations—possibly through improved training or decoding constraints—as a promising direction for future work.
>
> [7]Nayab S, Rossolini G, Simoni M, et al. Concise thoughts: Impact of output length on llm reasoning and cost[J]. arXiv preprint arXiv:2407.19825, 2024.
>
> [8]Lee A, Che E, Peng T. How well do llms compress their own chain-of-thought? a token complexity approach[J]. arXiv preprint arXiv:2503.01141, 2025.
>
> [9]Han T, Wang Z, Fang C, et al. Token-budget-aware llm reasoning[C]//Findings of the Association for Computational Linguistics: ACL 2025. 2025: 24842-24855.
>
> [10]Yang W, Ma S, Lin Y, et al. Towards thinking-optimal scaling of test-time compute for llm reasoning[J]. arXiv preprint arXiv:2502.18080, 2025.
>
> # Q10: Behavior under smaller user budgets & usefulness of predicted budgets
>
> Thank you for the question. As discussed in Appendix C, when the user specifies a budget smaller than the model’s predicted requirement, the model is able to compress its reasoning to a certain degree, though this ability is limited—large downward adjustments often lead to incomplete reasoning or quality degradation, which reflects a general limitation of current LLMs’ controllable-length generation.
>
> The predicted budget serves a practical purpose beyond functioning as a simple progress bar. By giving users an upfront estimate of the expected token cost or reasoning time, it enables informed decisions such as whether to proceed, terminate early, or switch to a faster model variant (e.g., GPT Fast vs. Thinking). In this way, the predicted budget provides actionable guidance for balancing cost, latency, and answer quality.

---

### Official Review · Reviewer_nvPF · 2025-11-03

**Soundness:** 3
**Presentation:** 3
**Contribution:** 3
**Rating:** 6
**Confidence:** 2

**Summary:**

This paper tackles the problem of overthinking in reasoning-focused LLMs, where models use unnecessarily long chains of thought for simple problems. The proposed SelfBudgeter framework adds a “token budgeting” mechanism that predicts an appropriate reasoning length before generating an answer. The system is trained in two phases: (1) a cold-start stage where the model learns to output both a token budget and a solution in a new structured format, and (2) a reinforcement learning (GRPO) stage using a reward that balances correctness, adherence to budget, and efficiency. Experiments on GSM8K, MATH500, and AIME2025 show that SelfBudgeter reduces response length by up to 61% (1.5B) and 48% (7B) while maintaining near-original accuracy.

**Strengths:**

1. Well-motivated: Addresses a clear inefficiency in current reasoning models (overlong CoTs) and ties the issue to both compute and user experience.
2. Simple, clear, and novel methodology: The dual-phase training and structured output (<budget>, <solution>) are intuitively designed and implemented. The PreB (Precise Budget Control) reward combines cosine shaping with tolerance margins to enforce tight adherence to predicted token budgets, which is conceptually sound and well-explained.
3. Quantitative impact: Strong empirical results demonstrating large reductions in token usage without accuracy loss; comparisons across multiple datasets and model sizes strengthen claims.

**Weaknesses:**

1. Limited baselines: Evaluation compares mostly against L1 and E1 methods; omits other adaptive compute or early-exit frameworks (e.g., speculative reasoning, dynamic stopping).
2. Ablation coverage: Lacks systematic analysis of individual reward components (budget penalty vs. PreB vs. correctness term) and their effect on efficiency and accuracy.
3. Limited experiments: Non-math benchmarks are lacking, albeit the baselines are trained specifically for math. Also while the paper highlights user-specified budgets, empirical evaluation of human-in-the-loop or interactive scenarios is limited.

**Questions:**

1. Does SelfBudgeter generalize to non-math reasoning tasks (e.g., code or commonsense)?
2. Could the authors clarify whether the model ever violates user-provided budgets during inference, and if so, how often?
3. How does the approach interact with speculative decoding or routing frameworks? Could SelfBudgeter be combined with those for further efficiency?
4. How sensitive are the results to the α parameter schedule beyond the linear decay? Would nonlinear or task-specific schedules yield further gains?

---

> ### Author Response · Authors · 2025-12-02
>
> We appreciate the reviewers’ positive assessment of our work. The reviewers highlight that our paper is well-motivated—addressing inefficiencies caused by overlong chains-of-thought and connecting them to both compute cost and user experience. They also note the simplicity and novelty of our approach, emphasizing the intuitive dual-phase training design, structured outputs, and the sound formulation of the PreB reward for precise budget control. Finally, the reviewers recognize the strong quantitative impact of our method, with substantial token-length reductions achieved without sacrificing accuracy across diverse datasets and model scales.
>
> # W1: Unintroduced adaptive-compute or early-exit methods
>
> The baselines mentioned by the reviewer focus on objectives and mechanisms that are not fully aligned with the problem we aim to address.
>
> To elaborate, speculative decoding [1] improves latency by using a draft model to propose multiple tokens per step. However, it does not reduce the number of generated tokens—the total reasoning length remains unchanged, and thus the method is orthogonal to our goal of learning to generate shorter, budget-aware reasoning traces. In fact, speculative decoding and our approach can be naturally combined: speculative decoding shortens wall-clock time per token, while SelfBudgeter shortens the number of tokens themselves, yielding complementary benefits.
>
> Similarly, dynamic stopping [2] halts decoding based on entropy or confidence signals, but it requires additional inference-time monitoring, which introduces overhead and complicates deployment. More importantly, dynamic stopping does not train the model to predict or internalize its own token budget. It acts as an external controller. In contrast, our method explicitly optimizes the model to learn its own token budget during training, integrating budget prediction into the reasoning process itself.
>
> For these reasons, we consider speculative decoding and dynamic stopping orthogonal rather than directly comparable. These techniques address inference-time acceleration, while our work focuses on teaching the model to estimate and adhere to a self-determined or user-specified budget, fundamentally targeting a different aspect of efficient reasoning. We appreciate the reviewer’s suggestions—if there are additional works the reviewer believes are directly comparable in terms of learned budget prediction, we would be grateful for clarification and are happy to follow up accordingly.
>
> [1]Pan R, Dai Y, Zhang Z, et al. Specreason: Fast and accurate inference-time compute via speculative reasoning[J]. arXiv preprint arXiv:2504.07891, 2025.
>
> [2]Yang C, Si Q, Duan Y, et al. Dynamic early exit in reasoning models[J]. arXiv preprint arXiv:2504.15895, 2025.
>
> # W2: Ablation of reward components
>
> Thank you for the question regarding the contribution of each reward component. We conducted a detailed ablation study by progressively adding format, correctness, preb, and budget-penalty terms. The results show that format-only training cannot preserve reasoning ability—accuracy collapses across all benchmarks. Adding correctness restores accuracy, but the model fails to achieve meaningful compression on harder tasks such as MATH500 and AIME2025. Incorporating preb helps stabilize budget prediction but hurts performance on easier tasks (e.g., GSM8K). Conversely, adding only the budget-penalty improves compression on easy problems but degrades accuracy on challenging ones. In contrast, SelfBudgeter, which combines all four components, consistently achieves the best accuracy–compression trade-off across datasets. These results indicate that the components are complementary, and only their joint use yields stable and reliable length control without sacrificing correctness. We have added the full analysis and additional examples to Appendix E.
>
> | Models / Reward              | GSM8K Acc | GSM8K Len | Math500 Acc | Math500 Len | AIME2025 Acc | AIME2025 Len |
> |----------------------|-----------|-----------|--------------|--------------|---------------|--------------|
> | R1-1.5B              | 73.09 ± 6.11 | 2865.08 ± 827.6 | 74.93 ± 2.31 | 5327.12 ± 299.3 | 22.22 ± 4.16 | 14444.03 ± 382.5 |
> | Format only          | 1.64 ± 1.36 | 4866.40 ± 660.2 | 1.47 ± 0.90 | 5558.35 ± 633.5 | 0.00 ± 0.00 | 6448.66 ± 1086 |
> | F+Correctness        | 84.58 ± 0.61 | 1258.45 ± 36.00 | 76.27 ± 1.33 | 3123.37 ± 85.86 | 20.00 ± 0.00 | 11095.39 ± 1188 |
> | F+C+Preb             | 82.89 ± 0.80 | 2151.26 ± 23.37 | 78.00 ± 0.86 | 3109.43 ± 38.03 | 15.56 ± 1.57 | 4671.86 ± 153.3 |
> | F+C+BudgetPenalty    | 84.26 ± 0.46 | 1263.38 ± 24.15 | 76.67 ± 0.19 | 2983.28 ± 16.03 | 21.11 ± 1.57 | 7263.38 ± 271.1 |
> | SelfBudgeter-1.5B    | 84.10 ± 0.38 | 1231.79 ± 15.95 | 78.47 ± 0.47 | 2326.85 ± 25.70 | 21.11 ± 1.57 | 4288.10 ± 130.2 |

---

> ### Author Response · Authors · 2025-12-02
>
> # W3-1 & Q1: Generalization ability
>
> Thank you for pointing this out—we agree that including non-math benchmarks helps present a more complete picture of model efficiency. In our original setup, we focused primarily on math reasoning tasks because overthinking phenomena are most prominently studied and systematically evaluated in math-oriented benchmarks, and prior adaptive-compute or cost-aware reasoning works have largely concentrated on this domain[3,4,5]. These tasks provide fine-grained control over reasoning length, making them suitable for analyzing budget allocation behavior.
>
> Nevertheless, we also evaluated our method on non-math reasoning datasets, and we now include those results in the table below for completeness. These results were generated as part of our earlier experimentation and are consistent with the trends observed on math benchmarks—SelfBudgeter reduces unnecessary reasoning tokens while preserving accuracy.
>
> | Models            | GPQA Acc      | GPQA Len           | KnowLogic Acc | KnowLogic Len        |
> |-------------------|---------------|---------------------|----------------|------------------------|
> | R1-1.5B           | 33.04 ± 2.28  | 11780.87 ± 928.3    | 10.14 ± 0.61   | 11695.94 ± 681.1      |
> | E1(0.5K,1K)       | 26.34 ± 1.28  | 1278.19 ± 31.92     | 6.69 ± 0.33    | 1272.04 ± 17.27       |
> | E1(4K,1K)         | 34.30 ± 2.38  | 2758.77 ± 82.87     | 12.23 ± 0.32   | 3327.37 ± 93.19       |
> | L1-Max(3600)      | 31.92 ± 0.79  | 3892.47 ± 797.3     | 13.69 ± 0.57   | 5145.91 ± 1291.41     |
> | SelfBudgeter-1.5B | 30.65 ± 0.74  | 3326.83 ± 25.93     | 16.26 ± 0.15   | 4491.35 ± 10.54       |
>
> [3]Chen X, Xu J, Liang T, et al. Do not think that much for 2+ 3=? on the overthinking of o1-like llms[J]. arXiv preprint arXiv:2412.21187, 2024.
>
> [4]Xu Y, Dong H, Wang L, et al. Scalable chain of thoughts via elastic reasoning[J]. arXiv preprint arXiv:2505.05315, 2025.
>
> [5]Aggarwal P, Welleck S. L1: Controlling how long a reasoning model thinks with reinforcement learning[J]. arXiv preprint arXiv:2503.04697, 2025.
>
> # W3-2 & Q2: Limited evaluation of user-specified budgets
>
> As noted in the paper, we analyze user-specified budget control in Appendix C. Specifically, we provide experiments where we manually set different target budgets and examine how the model adjusts its response length. The results show that the model’s output length decreases as the user-specified budget becomes smaller, though precise control becomes increasingly difficult once the budget is pushed to very low ranges.
>
> At the same time, we acknowledge that fully interactive or human-in-the-loop experiments would further enrich the study. A key challenge is that precise control of output length remains difficult for current LLMs, and aside from hard truncation, robust mechanisms for strict adherence are still limited across existing models[6,7,8,9].
>
> We plan to expand this direction in future work, particularly through more interactive settings and improved techniques for reliable budget adherence.
>
> [6]Nayab S, Rossolini G, Simoni M, et al. Concise thoughts: Impact of output length on llm reasoning and cost[J]. arXiv preprint arXiv:2407.19825, 2024.
>
> [7]Lee A, Che E, Peng T. How well do llms compress their own chain-of-thought? a token complexity approach[J]. arXiv preprint arXiv:2503.01141, 2025.
>
> [8]Han T, Wang Z, Fang C, et al. Token-budget-aware llm reasoning[C]//Findings of the Association for Computational Linguistics: ACL 2025. 2025: 24842-24855.
>
> [9]Yang W, Ma S, Lin Y, et al. Towards thinking-optimal scaling of test-time compute for llm reasoning[J]. arXiv preprint arXiv:2502.18080, 2025.
>
> # Q3: Compatibility with speculative decoding / routing frameworks
>
> In principle, SelfBudgeter is fully compatible with such systems: the learned budget-aware policy can serve as a drop-in replacement for the base model inside speculative decoding or routing pipelines. Because SelfBudgeter produces shorter reasoning traces, combining it with these methods can further reduce compute while preserving accuracy.
>
> More broadly, SelfBudgeter and system-level techniques address different parts of the efficiency pipeline. They can be used together rather than replacing one another.

---

> ### Author Response · Authors · 2025-12-02
>
> # Q4: Sensitivity to α-schedule choices
>
> Thank you for the question regarding the sensitivity of our method to different α-scheduling strategies. The additional experiments confirm that our conclusions remain robust: Linear scheduling consistently offers the most stable accuracy–compression trade-off, validating our choice.
>
> We provide additional experiments comparing three α-scheduling strategies (Fixed, Cosine, and Linear). In the Fixed setting, α is kept at 0.2, while in the Cosine and Linear settings α decreases from 6.0 to 0.1 following the formulas included in Appendix F. As shown in the table below, Fixed scheduling achieves the strongest compression but suffers from accuracy drops due to over-compression, with Cosine scheduling showing similar behavior. Linear scheduling provides a more favorable balance, maintaining accuracy while achieving over 61% length reduction. These analyses and formulas have been added to Appendix F of the revised manuscript.
>
> | α Scheduling        | GSM8K Acc     | GSM8K Len        | Math500 Acc   | Math500 Len       | AIME2025 Acc   | AIME2025 Len        |
> |---------------------|---------------|-------------------|----------------|--------------------|-----------------|----------------------|
> | Fixed Scheduling    | 79.88 ± 0.001 | 709.49 ± 5.987    | 73.06 ± 0.004 | 1015.96 ± 4.911    | 10.00 ± 0.047   | 1545.50 ± 61.72      |
> | Cosine Scheduling   | 83.22 ± 0.68  | 1149.43 ± 7.84    | 75.93 ± 1.61  | 1769.06 ± 24.35    | 17.78 ± 4.16    | 2366.34 ± 40.32      |
> | Linear Scheduling   | 84.10 ± 0.38  | 1231.79 ± 15.95   | 78.47 ± 0.47  | 2326.85 ± 25.70    | 21.11 ± 1.57    | 4288.10 ± 130.2      |

---

### Note · Authors · 2025-12-03

I have read and agree with the venue's withdrawal policy on behalf of myself and my co-authors.